# Tachykinin signaling inhibits task-specific behavioral responsiveness in honeybee workers

Bin Han[1,2], Qiaohong Wei[1], Fan Wu[1,3], Han Hu[1], Chuan Ma[1], Lifeng Meng[1], Xufeng Zhang[1,4], Mao Feng[1], Yu Fang[1], Olav Rueppell[2,5]*, Jianke Li[1]*

[1]Institute of Apicultural Research/Key Laboratory of Pollinating Insect Biology, Ministry of Agriculture, Chinese Academy of Agricultural Science, Beijing, China; [2]Department of Biology, University of North Carolina Greensboro, Greensboro, United States; [3]Biometrology and Inspection & Quarantine, College of Life Science, China Jiliang University, Hangzhou, China; [4]Institute of Horticultural Research, Shanxi Academy of Agricultural Sciences, Shanxi Agricultural University, Taiyuan, China; [5]Department of Biological Sciences, University of Alberta, Edmonton, Canada

**Abstract** Behavioral specialization is key to the success of social insects and leads to division of labor among colony members. Response thresholds to task-specific stimuli are thought to proximally regulate behavioral specialization, but their neurobiological regulation is complex and not well understood. Here, we show that response thresholds to task-relevant stimuli correspond to the specialization of three behavioral phenotypes of honeybee workers in the well-studied and important *Apis mellifera* and *Apis cerana*. Quantitative neuropeptidome comparisons suggest two tachykinin-related peptides (TRP2 and TRP3) as candidates for the modification of these response thresholds. Based on our characterization of their receptor binding and downstream signaling, we confirm a functional role of tachykinin signaling in regulating specific responsiveness of honeybee workers: TRP2 injection and RNAi-mediated downregulation cause consistent, opposite effects on responsiveness to task-specific stimuli of each behaviorally specialized phenotype but not to stimuli that are unrelated to their tasks. Thus, our study demonstrates that TRP signaling regulates the degree of task-specific responsiveness of specialized honeybee workers and may control the context specificity of behavior in animals more generally.

*For correspondence:
olav@ualberta.ca (OR);
apislijk@126.com (JL)

**Competing interests:** The authors declare that no competing interests exist.

## Introduction

Behavioral responses of animals to external and internal stimuli have evolved to optimize survival and reproduction under average circumstances (*Darwin, 1859*). However, environmental and interindividual variability commonly causes deviations from the average, resulting in selection for context-specific and condition-dependent behavior (*Dall et al., 2004*; *Bolnick et al., 2003*; *West-Eberhard, 1989*). Such plasticity is limited and evolutionary constraint (*Arnold, 1992*) of behavior occurs in the form of behavioral syndromes: differences among individuals that manifest across different contexts (*Sih et al., 2004*). Advantages of behavioral plasticity and specificity have been documented in many systems and some neuroendocrine mechanisms have been identified (*O'Connell and Hofmann, 2011*; *Kim et al., 2017*). However, neural mechanisms that allow the sophistication of behavioral repertoires by regulating the context specificity of behavioral responses remain insufficiently understood.

Behavioral regulation is particularly important in social species in which social interactions provide a high diversity of behavioral context (*Oliveira, 2009*; *Gronenberg and Riveros, 2009*). Social

evolution also allows individuals to restrict their behavioral repertoires through temporal or permanent behavioral specialization (*Oster and Wilson, 1978*). This specialization and the resulting division of labor are believed to be major contributors to the successful colony life of many social insects despite its potential costs (*Jeanson and Weidenmüller, 2014*). Advanced social evolution thus allows interindividual plasticity to replace individual behavioral plasticity. Nevertheless, the principal problem of behavioral plasticity across different contexts remains the same, and social insects can be constrained in their behavioral evolution by correlated selection responses across different behaviors or castes (*Rueppell et al., 2006*; *Jandt et al., 2014*).

Behavior often occurs in response to a specific stimulus exceeding an individual's specific response threshold (*Tinbergen, 1951*; *Mayr, 1974*). Response thresholds depend on internal physiological states (*Ricklefs and Wikelski, 2002*) and translate the value of perceived stimuli into probabilities of behavioral responses and vary among individuals (*Scheiner et al., 2004*). In social insects, individual variation in response thresholds is linked to division of labor (*Theraulaz et al., 1998*; *Beshers et al., 1999*; *Robinson, 1992*; *Schulz et al., 2002*), and numerous studies have characterized this link across multiple levels of biological organization (*Scheiner et al., 2004*; *Page et al., 2012*; *Johnson, 2010*). The Western honeybee (*Apis mellifera*) is the most-studied social insect model. Many aspects of its division of labor are driven by a lifelong behavioral ontogeny, leading to age polyethism (*Seeley, 1982*). Young bees perform numerous inside tasks, most prominently brood care in form of alloparental nursing behavior, before transitioning to a mix of other in-hive tasks (*Johnson, 2008*). Similar to the highly specialized nursing stage, the final behavioral state of older bees as outside foragers is almost exclusive of other tasks (*Seeley, 1982*). Moreover, foragers often specialize on collecting only one of the principal food sources, pollen or nectar (*Page et al., 2000*). Specialized nurse bees (NBs), nectar foragers (NFs), and pollen foragers (PFs) are also found in the closely related Eastern honeybee, *Apis cerana* (*Ji et al., 2020*), thus offering a suitable parallel system for our study.

The behavioral specialists of NBs, NFs, and PFs exhibit pronounced differences in their responsiveness to task-related stimuli. Responsiveness to brood pheromones peaks at typical nursing age (*Pankiw, 2004*). In contrast, foragers have a lower response threshold to sugars and light than nurses (*Değirmenci et al., 2018*; *Ben-Shahar et al., 2003*). Among foragers, pollen specialists exhibit higher responsiveness to sucrose and pollen stimuli than NFs (*Page et al., 1998*; *Pankiw and Page, 1999*). Response thresholds can be quantified based on the honeybees' reflexive extension of their proboscis in response to stimuli, such as sucrose (*Scheiner et al., 2004*). The spontaneous proboscis extension reflex (PER) to sucrose has been expanded to other stimuli that bees spontaneously respond to *Nicholls and de Ibarra, 2013*; *Zhang et al., 2020* and conditioned stimuli to which no spontaneous responses occur (*Giurfa and Sandoz, 2012*).

Response thresholds can be modified by biogenic amines, and dopamine, 5-hydroxy-tryptamine, octopamine, and tyramine have been implicated in the regulation of different behaviors of worker bees although the mechanisms are not entirely clear (*Schulz et al., 2002*; *Scheiner et al., 2006*). Few studies have addressed the role of neuropeptides although they are a diverse group of neurotransmitters that can also act as neurohormones on distal targets to coordinate a wide range of internal states and behavioral processes (*Nässel, 2002*). Neuropeptides are intimately involved in food perception and social interaction of insects (*Schoofs et al., 2017*), two processes that are central to division of labor in social insects (*Ament et al., 2010*). Neuropeptides mediate pheromonal effects on physiology (*Shankar et al., 2015*; *Gendron et al., 2014*) and usually exhibit a high degree of specificity (*Inagaki et al., 2014*; *Taghert and Nitabach, 2012*). Therefore, neuropeptides are prime candidates for mediating the independent adjustment of socially relevant response thresholds and they have been implicated in honeybee worker specialization and division of labor (*Ji et al., 2020*; *Brockmann et al., 2009*).

More than 100 mature neuropeptides derived from 22 protein precursors have been identified in the Western honeybee, *A. mellifera* (*Han et al., 2015*; *Boerjan et al., 2010*). Several neuropeptides, including allatostatin, leucokinin, and tachykinin-related peptides (TRPs), may be involved in the control of social behavior of *A. mellifera* and the closely related *A. cerana*, such as aggression (*Pratavieira et al., 2018*), foraging (*Ji et al., 2020*; *Brockmann et al., 2009*), brood care (*Han et al., 2015*), and possibly a wide array of other behaviors (*Pratavieira et al., 2014*). However, most of these results are based on correlations between behavior and neuropeptide expression, and more

detailed studies are needed to understand the causal roles of neuropeptides in the behavioral specialization among honeybee workers.

Here, we report the results of a comprehensive study to test the hypothesis that neuropeptides regulate the division of labor in honeybees. We initially compared response thresholds to task-relevant stimuli among behaviorally defined worker groups (NBs, NFs, PFs) of the two closely related honeybee species *A. mellifera* and *A. cerana*. These response thresholds were correlated with neuropeptide expression levels, especially TRPs, suggesting a role of TRPs in worker specialization. Based on these results, we demonstrated in a series of TRP injections and RNAi-mediated knockdown of the *TRP* and its receptor (*TRPR*) a causal role of this pathway in modulating different response thresholds in a task-specific manner. We characterized the TRP signaling pathway molecularly by studying the TRPR properties and the important downstream events of cAMP and calcium accumulation and extracellular-signal-regulated kinase (ERK) activation in cell culture. Finally, we confirmed this connection in vivo by demonstrating corresponding effects of TRP injections and RNAi-mediated knockdown of *TRP* and *TRPR* on ERK phosphorylation.

## Results

### The task-specific responsiveness of worker bees showed significant variations between behavioral phenotypes and the two honeybee species

Based on the response threshold model for division of labor, we initially studied the responsiveness of three different behavioral phenotypes to three different task-relevant stimuli in *A. mellifera* and *A. cerana* using the PER assay. In accordance with our predictions, we identified significant differences among worker behavioral phenotypes in the PER responsiveness to the task-specific stimuli, including sucrose solution, pollen, and larva (*Figure 1*, *Figure 1—source data 1*, *Supplementary file 1*).

The percentage of bees showing a PER increased with sucrose concentration across all experimental groups (*Figure 1A*). In both, *Apis mellifera ligustica* (AML) and *Apis cerana cerana* (ACC), the sucrose response scores (SRSs) of PFs were higher than the SRSs of NFs (AML: $Z = 7.0$, p$\leq$0.001; ACC: $Z = 6.1$, p<0.001) and nurse bees (NBs) (AML: $Z = 5.9$, p<0.001; ACC: $Z = 5.2$, p<0.001), while no significant difference between NFs and NBs was observed in either species (*Figure 1B*). PFs were more responsive than NFs and NBs to all sucrose concentrations. The species comparison between AML and ACC showed significant higher sucrose responsiveness in PFs of AML than in PFs of ACC ($Z = 2.361$, p=0.018), specifically at sucrose concentrations of 0.3% ($\chi^2 = 4.1$, p=0.042), 1.0% ($\chi^2 = 5.2$, p=0.001), 3.0% ($\chi^2 = 8.4$, p=0.023), and 10.0% ($\chi^2 = 5.3$, p=0.021). NFs of AML and ACC showed no significant difference in overall SRS, but NFs of AML were more responsive than NFs of ACC at sucrose concentrations of 0.3% ($\chi^2 = 4.5$, p=0.035), 1.0% ($\chi^2 = 4.5$, p=0.033), and 3.0% ($\chi^2 = 4.0$, p=0.046). There was no significant difference between NBs of AML and ACC in sucrose responsiveness.

In AML, PFs were more responsive to pollen stimulation than NFs ($\chi^2 = 14.9$, p=0.002) and NBs ($\chi^2 = 20.2$, p<0.001), while there were no significantly statistical differences between NFs and NBs. Likewise, PFs of ACC were more sensitive than NFs ($\chi^2 = 6.0$, p=0.015) and NBs ($\chi^2 = 7.8$, p=0.001) without a statistically significant difference between NFs and NBs. PFs of AML showed a significant higher pollen responsiveness than of ACC ($\chi^2 = 4.9$, p=0.031), with no significant species differences in NFs and NBs (*Figure 1C*).

In larva responsiveness assay, NBs of AML showed increased responsiveness to larva stimulation compared to PFs ($\chi^2 = 7.2$, p=0.006) and NFs ($\chi^2 = 10.3$, p=0.001). Likewise, NBs of ACC were more sensitive than PFs ($\chi^2 = 4.2$, p=0.013) and NFs ($\chi^2 = 6.1$, p=0.002). NBs of AML were significantly more sensitive to larvae ($\chi^2 = 4.3$, p=0.027) than NBs of ACC, with no significant species differences in PFs and NFs (*Figure 1D*). In sum, our results indicated that behavioral specialization among the workers of both honeybee species correspond to differences in the workers' responsiveness to task-relevant stimuli. Correlative evidence linking different neuropeptides to foraging specialization in *A. mellifera* (*Pratavieira et al., 2018*) motivated us to further investigate whether brain neuropeptides could regulate the response thresholds.

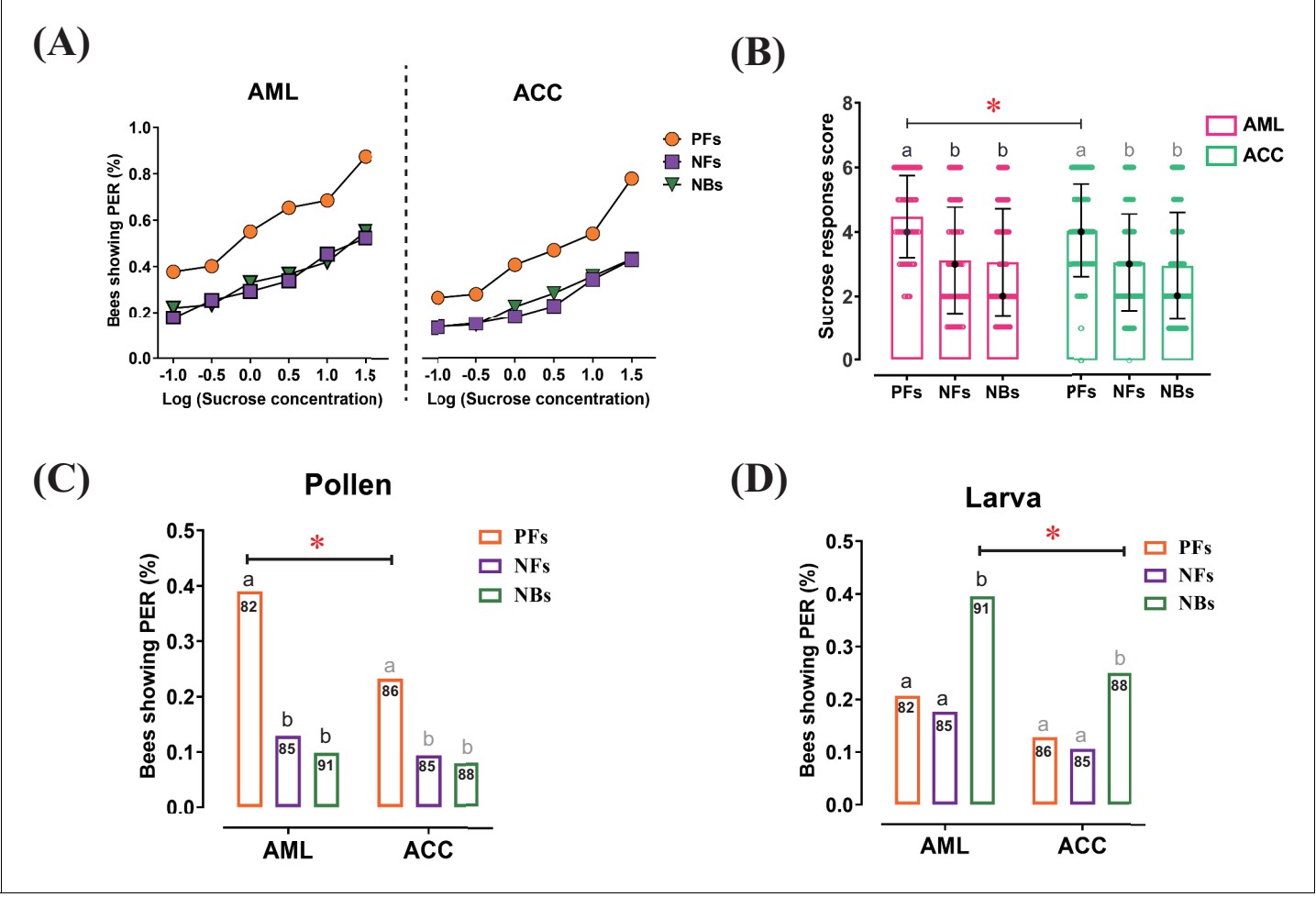

**Figure 1.** Responses to sucrose solution, pollen, and larva stimulations were significantly different among behavioral phenotypes and between honeybee species. (**A**) The proportion of pollen foragers (PFs), nectar foragers (NFs), and nurse bees (NBs) showing a proboscis extension reflex (PER) increased with increasing concentrations of sucrose solutions. Left: *Apis mellifera ligustica* (AML), right: *Apis cerana cerana* (ACC). Details of the statistical results of our comparisons of sucrose responsiveness between behavioral phenotypes and bee species are listed in ***Supplementary file 1***. (**B**) Median sucrose response scores (SRS; intermediate lines) and quartiles (upper and lower lines) of PFs, NFs, and NBs. The number of bees per group was between 125 and 136. Kruskal-Wallis tests with Bonferroni correction were used to compare the SRSs of the three behavioral phenotypes in the same species and significant differences are denoted by letters at p<0.05. Pairwise Mann-Whitney U tests were used for comparing the same phenotype between two honeybee species (*p<0.05). (**C**) Proportion of PFs, NFs, and NBs showing PER to pollen stimulation of their antennae. (**D**) Proportion of PFs, NFs, and NBs showing PER to antennal stimulation with larvae. Numbers in bars represent the number of individuals sampled in each group. Independent chi-square tests were used to compare the responsiveness to pollen or larvae between species (*p<0.05) and among behavioral phenotypes within species (letters indicate significant difference at p<0.05).

The online version of this article includes the following source data for figure 1:

**Source data 1.** The proboscis extension response of *Apis mellifera ligustica* and *Apis cerana cerana* worker bees to different sucrose solutions.

## Quantitative neuropeptidomics revealed consistent correlations of TRPs with behavior

To identify neuropeptides that potentially influence behavioral specialization, we compared the whole brain neuropeptidomes of NBs, PFs, and NFs of AML and ACC with liquid chromatography with tandem mass spectrometry (LC-MS/MS). The comparisons revealed numerous differences among experimental groups, but only two tachykinin-related peptides (TRP2 and TRP3) showed consistent patterns relating to the task-specific responsiveness of the experimental groups.

Overall, 132 unique neuropeptides derived from 23 neuropeptide families were identified in the brain of AML worker bees (***Supplementary file 2***). In the brain of ACC worker bees, 116 unique neuropeptides derived from 22 neuropeptide families were identified (***Supplementary file 3***).

Quantitative comparison among the three behavioral phenotypes of AML showed that 40 neuropeptides derived from 16 neuropeptide families were differentially expressed the brain (*Figure 2* and *Figure 2—source data 1*). Among 19 differentially expressed neuropeptides between PFs and NFs, nine neuropeptides were upregulated in PFs and 10 were upregulated in NFs. Among 24 differentially expressed neuropeptides between PFs and NBs, 18 were upregulated in PFs and six were upregulated in NBs. Moreover, 21 differentially expressed neuropeptides were found between NFs and NBs, with 14 upregulated in PFs and seven upregulated in NBs. In ACC, 18 neuropeptides were differentially expressed between PFs and NFs, with nine upregulated in each group. Between PFs and NBs, 27 neuropeptides showed different expression levels: 20 were upregulated in PFs and seven were upregulated in NBs (*Figure 2—source data 2*). Twenty-five neuropeptides were differentially expressed between NFs and NBs, with 19 upregulated in NFs and six in NBs. In the species comparison between AML and ACC, the number of differentially expressed neuropeptides in NBs, PFs, and NFs was 13, 10, and 11, of which 7, 6, and 6 were upregulated in AML, respectively (*Figure 2—source data 3*). Thus, this global comparison of brain neuropeptide levels revealed numerous quantitative and qualitative differences. However, consistent differences between NBs, NFs, and PFs of both species were only found for TRP2 and TRP3, prompting their further study.

## TRP signaling inhibited task-specific responsiveness

Based on our behavioral and peptidomics results, we hypothesized tachykinin signaling to affect response thresholds and tested this prediction with a combination of gain- and loss-of-function experiments. We focused on TRP2 because it showed a stronger binding affinity to the receptor TRPR than TRP3 in our biochemical work (see 'TRP/TRPR signaling was found to affect the $G_{\alpha q}$ and $G_{\alpha s}$ pathways and trigger the ERK cascade' section).

### TRP2 injection decreased task-specific responsiveness

Task-specific responsiveness of the different behavioral phenotypes (PFs, NFs, and NBs) was decreased by injection of TRP2 in a task-specific manner (*Figure 3* and *Figure 3—source data 1*).

Injection of the TRP2 peptide significantly reduced the SRS of PFs ($Z$ = 2.2, p=0.031), significantly reducing PER responses to all sucrose concentrations used. Similarly, NFs injected with TRP2 displayed significantly lower SRS than control-injected NFs ($Z$ = 2.3, p=0.019), significantly reducing PER responses to all sucrose concentrations except 0.1% (*Figure 3A and B*, *Figure 3—source data 1*). In contrast, TRP2-injected NBs did not show significant responsiveness changes to sucrose relative to controls. For pollen stimulation, PFs showed significantly decreased responsiveness to pollen loads after TRP2 injection ($\chi^2$ = 6.7, p=0.017), while no significant effects were observed in PFs and NFs (*Figure 3C*). In the larval responsiveness assay, injection of TRP2 only significantly affected the responsiveness of NBs ($\chi^2$ = 6.1, p=0.001) but not NFs or PFs (*Figure 3D*).

### Downregulation of *TRP* or *TRPR* increased task-specific responsiveness

The function of TRP/TRPR signaling on task-specific responsiveness was further confirmed by RNAi-mediated downregulation of *TRP* or *TRPR* that increased task-specific responsiveness, opposite to the results of the TRP2 injection.

Knockdown efficiencies were close to 60% for *TRP* and *TRPR* mRNA levels at 24 hr post-injection of the corresponding dsRNA (*Supplementary file 4*). Therefore, subsequent PER assays were performed 24 hr after dsRNA injection. Relative to control injections, knockdown of either *TRP* or *TRPR* significantly increased the SRS of NFs (ds*TRP*: $Z$ = 2.4, p=0.049; ds*TRPR*: $Z$ = 2.6, p=0.025), specifically increasing the responses of NFs to sucrose at concentrations of 0.1% (ds*TRP*:$\chi^2$ = 3.9, p=0.039; ds*TRPR*: $\chi^2$ = 4.9, p=0.023), 0.3% (ds*TRP*: $\chi^2$ = 5.3, p=0.018; ds*TRPR*: $\chi^2$ = 4.3, p=0.030), 1.0% (ds*TRP*: $\chi^2$ = 7.0, p=0.007; ds*TRPR*: $\chi^2$ = 6.6, p=0.009), and 3.0% (ds*TRP*: $\chi^2$ = 6.0, p=0.012; ds*TRPR*: $\chi^2$ = 7.4, p=0.006) (*Figure 4A and B*, *Figure 4—source data 1*, *Supplementary file 5*). Knockdown of *TRP* and *TRPR* did not significantly change the overall SRS of PFs and NBs, although it significantly increased the responses of PFs to sucrose at concentrations of 0.1% (ds*TRP*: $\chi^2$ = 4.4, p=0.029; ds*TRPR*: $\chi^2$ = 6.1, p=0.011), 0.3% (ds*TRP*: $\chi^2$ = 5.2, p=0.018; ds*TRPR*: $\chi^2$ = 6.0, p=0.011), and 1.0% (ds*TRP*: $\chi^2$ = 5.0, p=0.020; ds*TRPR*: $\chi^2$ = 4.7, p=0.025). Responses to pollen stimulation after dsRNA injection indicated that knockdown of either *TRP* or *TRPR* specifically increased the pollen responsiveness of PFs (ds*TRP*: $\chi^2$ = 6.5, p=0.018; ds*TRPR*: $\chi^2$ = 6.4, p=0.010), whereas the

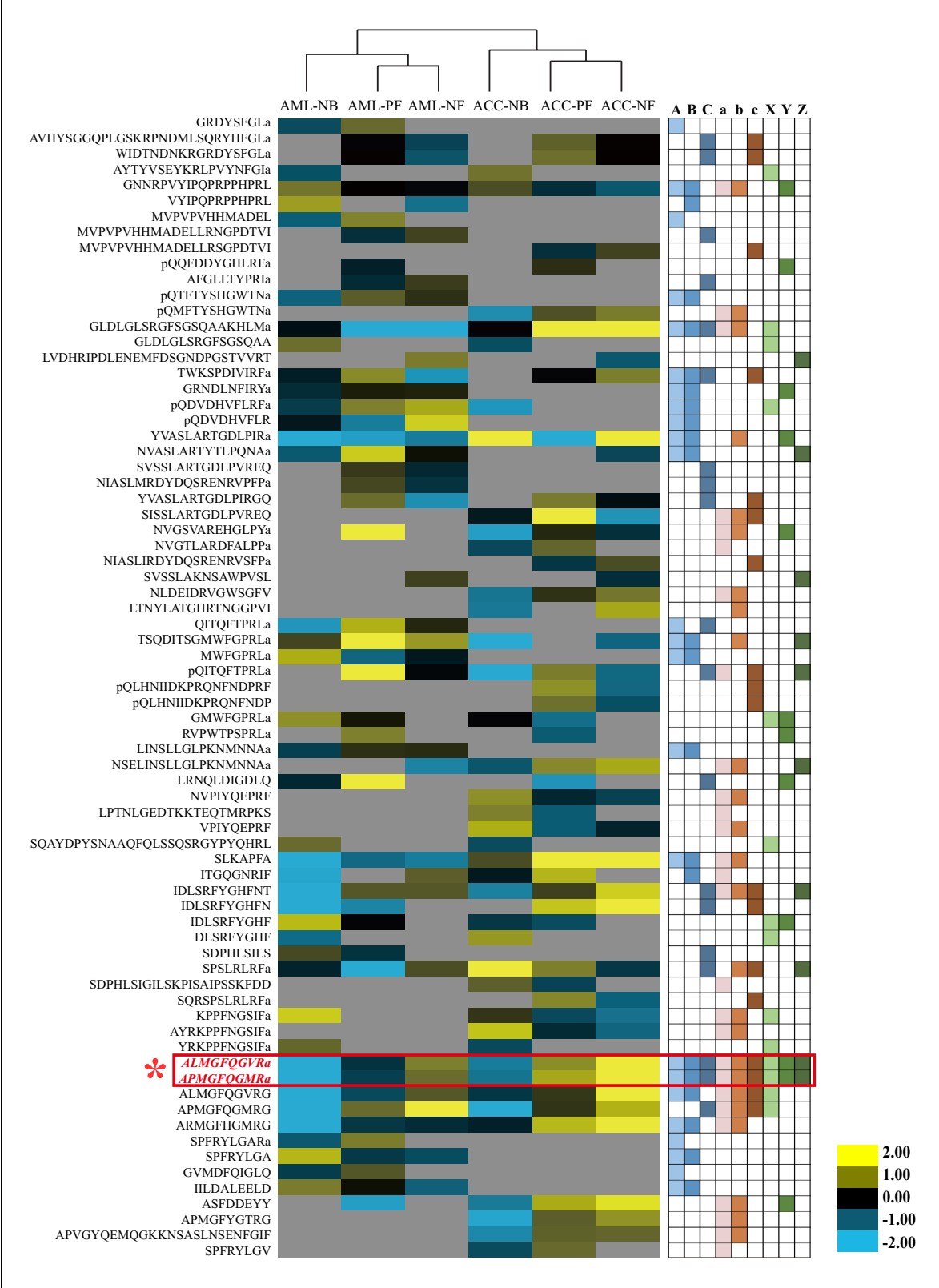

**Figure 2.** Quantitative comparison of the brain neuropeptides among behavioral phenotypes and species. The brain neuropeptides were quantitatively compared between nurse bees (NBs), pollen foragers (PFs), and nectar foragers (NFs) of *Apis mellifera ligustica* (AML) and *Apis cerana cerana* (ACC). The up- and downregulated peptides are indicated by yellow and blue colors, respectively. Color intensity indicates the relative expressional level, as noted in the key. Letters A, B, and C on the right represent significant differences between NBs and PFs, NBs and NFs, and PFs and NFs in AML,

*Figure 2 continued on next page*

*Figure 2 continued*

respectively; a, b, and c represent significant differences between NBs and PFs, NBs and NFs, and PFs and NFs in ACC, respectively; X, Y, and Z represent significant differences of NBs, PFs, and NFs between AML and ACC, respectively. For detailed quantitative comparison results, see *Figure 2—source data 1*, *2,* and *3*.

The online version of this article includes the following source data for figure 2:

**Source data 1.** Quantitative neuropeptide comparison of different behavioral phenotypes of *Apis mellifera ligustica* workers.
**Source data 2.** Quantitative neuropeptide comparison of different behavioral phenotypes of *Apis cerana cerana* workers.
**Source data 3.** Quantitative neuropeptide comparison between *Apis cerana cerana* and *Apis mellifera ligustica*.

effects on NFs and NBs were not significant (*Figure 4C*). The responsiveness of NBs to larvae was significantly increased after gene knockdown of either *TRP* ($\chi^2$ = 4.4, p=0.029) or *TRPR* ($\chi^2$ = 4.8, p=0.023), but NFs and PFs were not affected (*Figure 4D*).

The experimental up- and downregulation of TRP signaling resulted in complementary changes in the responsiveness of our experimental groups in a specific way: Only the particular, increased

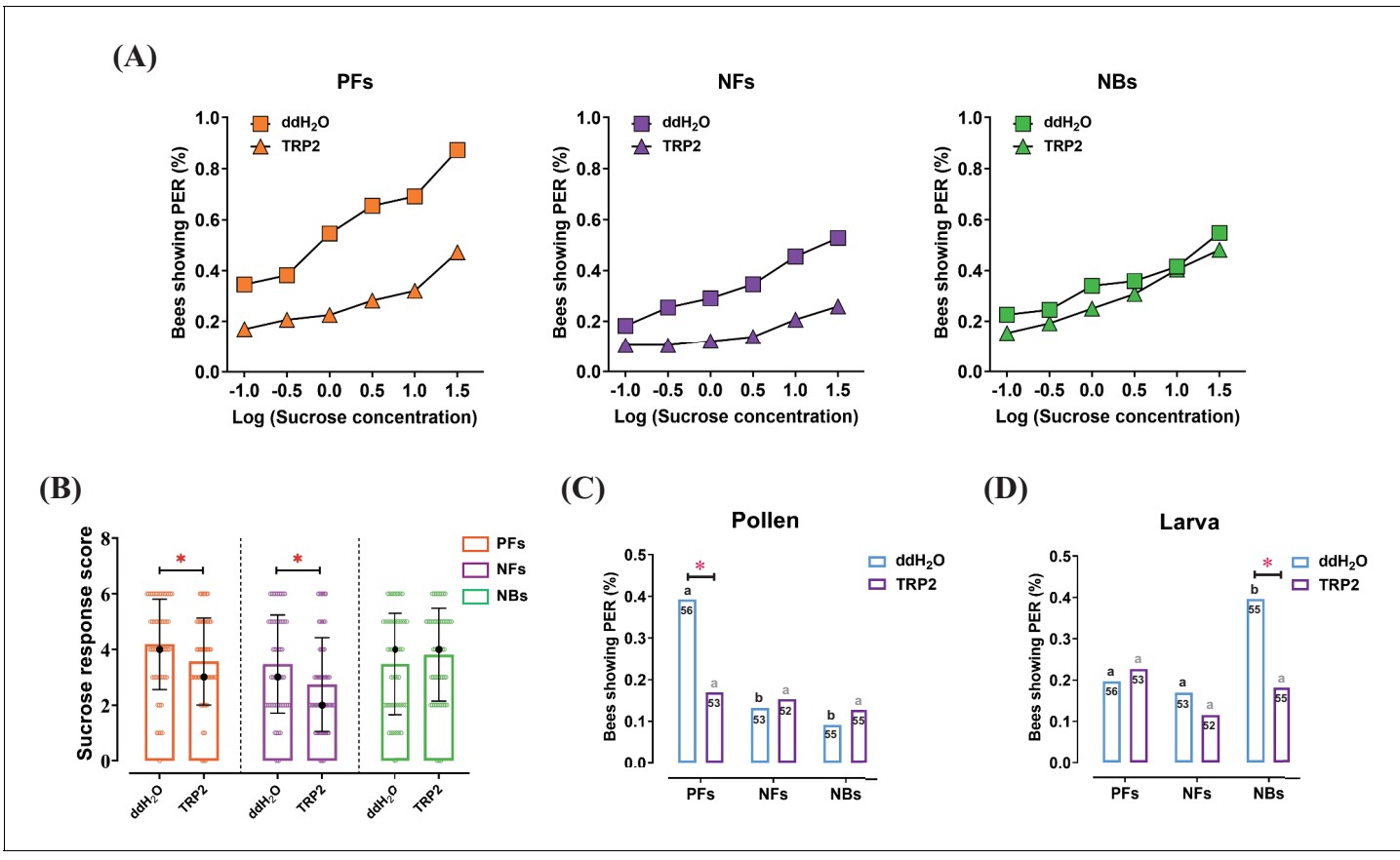

**Figure 3.** Injection of tachykinin-related peptide 2 (TRP2) decreased task-specific responsiveness of worker bees ( *Apis mellifera ligustica*). (**A**) The proportion of pollen foragers (PFs), nectar foragers (NFs), and nurse bees (NBs) exhibiting a positive proboscis extension reflex (PER) increased with increasing concentrations of sucrose solutions but was overall decreased in PFs and NFs after injection of TRP2 compared to ddH₂O injection. (**B**) Median sucrose response scores (SRS; intermediate lines) and quartiles (upper and lower lines) of ddH₂O-injected and TRP2-injected groups of PFs, NFs, and NBs. The number of bees varied between 52 and 58 per group. Mann-Whitney U tests were used to compare the SRS (*p<0.05). The proportion of PFs, NFs, and NBs showing PER to pollen stimulation (**C**) and larva stimulation (**D**) after injection of TRP2 or ddH₂O. Numbers in bars are the number of individuals sampled in each group. Independent chi-square tests were used to compare the responsiveness between different treatments (*p<0.05) and between different behavioral phenotypes within treatments (significant differences are denoted by letters, p<0.05).

The online version of this article includes the following source data for figure 3:

**Source data 1.** The proboscis extension response of workers after injection of ddH₂O and tachykinin-related peptide 2 (TRP2).

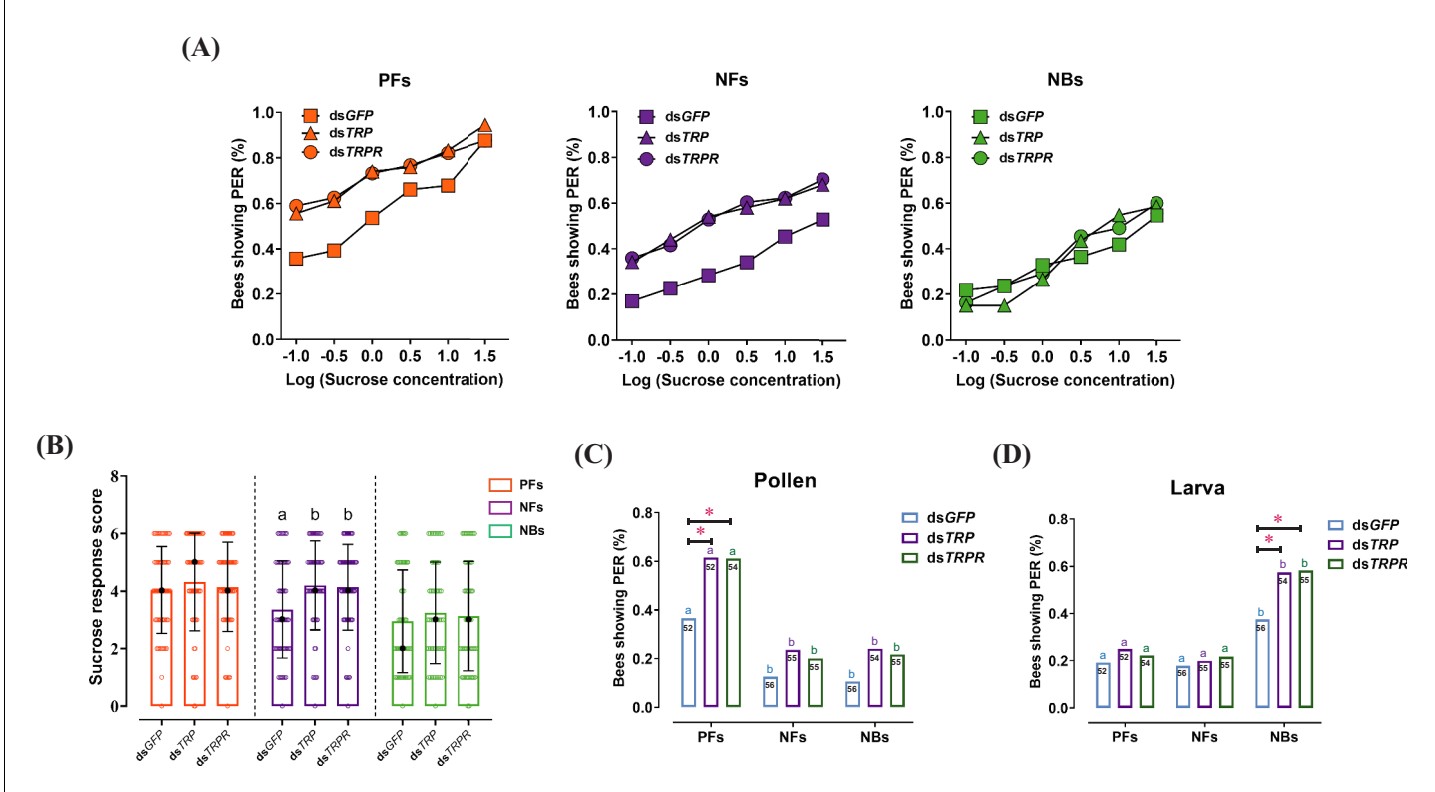

**Figure 4.** RNAi-mediated knockdown of tachykinin-related peptide (*TRP*) and its receptor (*TRPR*) expression increased task-specific responses of worker bees ( *Apis mellifera ligustica*). (A) Proportion of positive proboscis extension reflex (PER) responses of pollen foragers (PFs), nectar foragers (NFs), and nurse bees (NBs) increases with increasing concentrations of sucrose solutions but overall increases occur only in PFs and NFs after knockdown of *TRP* or *TRPR* transcripts compared to GFP control. Statistical details of these sucrose responsiveness comparisons are shown in *Supplementary file 5*. (B) Median sucrose response scores (SRS; intermediate lines) and quartiles (upper and lower lines) of ddH$_2$O-injected and TRP2-injected PFs, NFs, and NBs. The number of bees per group varied between 50 and 56. Kruskal-Wallis tests with Bonferroni correction were used to compare the SRSs of the three treatment groups of each behavioral phenotype and significant differences are denoted by letters (p<0.05). The proportion of PFs, NFs, and NBs showing PER to pollen stimulation (C) and larvae stimulation (D) after *GFP*, *TRP*, or *TRPR* knockdown. Numbers in bars are the number of individuals sampled in each group. Independent chi-square tests were used to compare the task-specific responsiveness between different treatments (*p<0.05, **p<0.01) within behavioral phenotypes and between different behavioral phenotypes within each treatment (significant differences are denoted by letters, p<0.05).

The online version of this article includes the following source data for figure 4:

**Source data 1.** The proboscis extension response of workers after injection of ds*GFP*, ds*TRP*, and ds*TRPR*.

responsiveness to task-relevant stimuli of each behavioral group was affected, suggesting that TRP signaling generally regulates the degree of behavioral specialization by moderating the responsiveness to task-specific stimuli in behavioral specialists.

## TRP/TRPR signaling was found to affect the G$_{αq}$ and G$_{αs}$ pathways and trigger the ERK cascade

The action of most insect neuropeptides is mediated by G-protein-coupled receptors (GPCRs) that activate cAMP- and Ca$^{2+}$-dependent pathways, such as ERK signaling to affect diverse biological processes (*Hauser et al., 2006*; *He et al., 2014*; *Werry et al., 2005*). However, the details of TRP signaling vary among insects (*Birse et al., 2006*; *Poels et al., 2007*) and have not been well studied in honeybees. To further support our behavioral studies by providing a plausible biochemical action of TRP signaling, we thus confirmed that the honeybee TRPR was localized in the cell membrane and specifically activated by TRP, triggering intracellular cAMP accumulation, Ca$^{2+}$ mobilization, and ERK phosphorylation by dually coupling G$_{αs}$ and G$_{αq}$ signaling pathways. Because TRP2 and TRP3 are very similar and TRP2 displayed slightly higher binding affinity to their common receptor (see below), we used TRP2 only in a few of the below experiments.

The honeybee *TRPR* gene was cloned and expressed in the human embryonic kidney cells (HEK293) and the insect *Spodoptera frugiperda* pupal ovary cells (Sf21). Significant cell surface expression was observed by fluorescence microscopy (*Figure 5A* and *Figure 5—figure supplement 1*), revealing that the honeybee TRPR was exclusively localized in the cell membrane. Competitive binding assays with labeled TRP2 and TRP3 confirmed the predicted high affinity of the TRPR for both, although it was higher for TRP2 than for TRP3 (*Figure 5—figure supplement 1*, *Figure 5—source data 1*).

The detected accumulation of intracellular cAMP concentration only in HEK293 cells transformed with TRPR (*Figure 5B*, *Figure 5—source data 2*) confirmed that TRP2 and TRP3 can activate TRPR and trigger cAMP signaling. This effect was confirmed for both cell types in further dose-response experiments and compared to other neuropeptides, including short neuropeptide F (SNF), pigment-dispersing hormone (PDH), and corazonin (CRZ), which did not induce any detectable cAMP accumulation (*Figure 5—figure supplement 2*, *Figure 5—source data 2*). Correspondingly, selective inhibition/activation experiments implicated the $G_{\alpha s}$ and $G_{\alpha q}$ (but not $G_{\alpha i}$) subunits in this signaling mechanism (*Figure 5—figure supplement 3*, *Figure 5—source data 3*). Furthermore, intracellular $Ca^{2+}$ mobilization was also found to result from TRP2 or TRP3 stimulation, dependent on $G_{\alpha q}$ (*Figure 5C*, *Figure 5—figure supplement 4*, and *Figure 5—source data 4*).

The final experiment of our in vitro studies demonstrated ERK phosphorylation in response to TRP/TRPR signaling by Western blot analysis. Treatment with different concentrations of TRP2 induced a transient, dose-dependent phosphorylation of ERK in both HEK293 ($EC_{50}$ = 68.04 nM) and Sf21 ($EC_{50}$ = 1.68 nM) cells (*Figure 5D*, *Figure 5—figure supplement 5*, *Figure 5—source data 5*). Further time-dependent analysis indicated that TRP2 elicited transient phosphorylation of ERK with maximal phosphorylation at 2 min and near basal levels by 90 min (*Figure 5—figure supplement 5*, *Figure 5—source data 5*). Moreover, specific inhibitors were used to elucidate TRP/TRPR signaling-mediated ERK activation in both HEK293 and Sf21 cells. Treatment with MEK inhibitor U0126, PKA inhibitor H89, and PKC inhibitor Go6983, respectively, led to a significant inhibition of TRP/TRPR-mediated ERK activation, whereas $G_{\alpha i}$ inhibitor pertussis toxin (PTX) had no effect, demonstrating that honeybee TRP/TRPR signaling dually coupled to $G_{\alpha s}$ and $G_{\alpha q}$ proteins to activate the ERK signaling pathway (*Figure 5—figure supplement 5*, *Figure 5—source data 5*).

## Regulation of ERK activity by TRP/TRPR signaling was confirmed in vivo

To complement our finding that TRP/TRPR signaling activates ERK phosphorylation in cell culture, we used our in vivo manipulations of TRP signaling to confirm the link between TRP and ERK signaling in living honeybee workers. Western blot results confirmed that TRP/TRPR signaling triggers ERK signaling in vivo. The level of phosphorylated ERK significantly increased after injection of TRP2 peptide into NBs, PFs, and NFs (*Figure 6A*) and decreased after knockdown of the *TRP* or *TRPR* transcripts (*Figure 6B*, *Figure 6—source data 1*).

## Discussion

Behavioral plasticity plays a central role in animal adaptation and modulating behavioral responsiveness to different stimuli and contexts is key to individual fitness. The success of social insects is partly due to their efficient division of labor, a form of behavioral plasticity among instead of within individuals. In this study, we demonstrated that the responsiveness to task-relevant stimuli correlates with behavioral specialization in two different honeybee species. Through parallel characterization of the neuropeptidome, we identified two tachykinin-related peptides (TRP2 and TRP3) as putative mechanism to adjust task-specific response thresholds and thus proximally guide division of labor. Subsequently, we characterized the molecular action of TRP2 and TRP3 in cell culture by verifying their binding to their membrane-bound receptor and demonstrating activation of multiple downstream signaling mechanisms. Finally, we verified causal involvement of TRP signaling in modulating task-specific behavioral response thresholds through complementary outcomes of TRP2 injection and RNAi-mediated knockdown of *TRP* and *TRPR*: while injection decreased task-specific responses, downregulation of TRP or TRPR increased the same specific responses. Thus, we present a mechanism that tunes the behavioral responsiveness of animals to specific stimuli compared to others. We use behaviorally specialized honeybee workers as models but hypothesize that this function of TRP

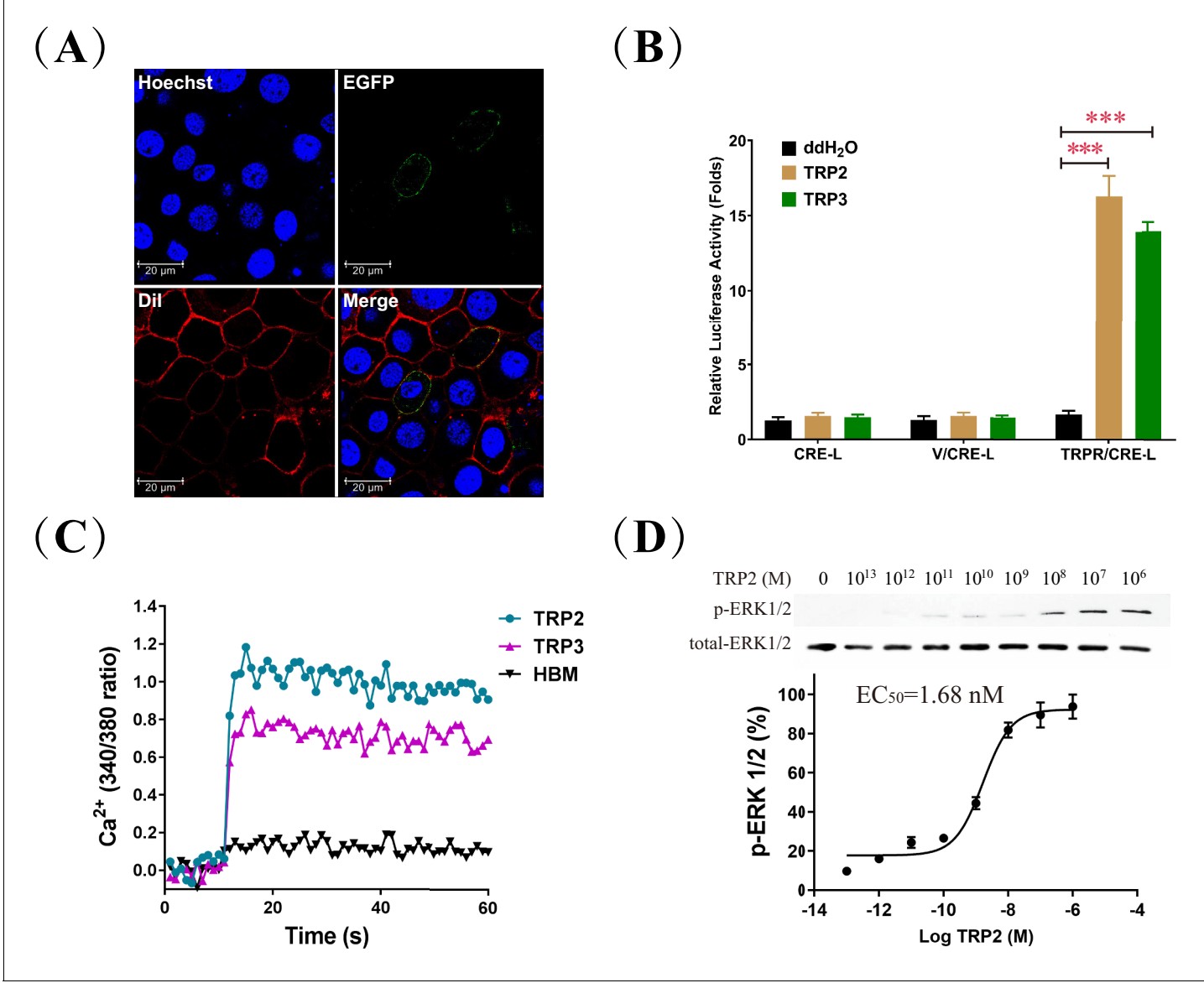

**Figure 5.** Biochemical characterization of *Apis mellifera* tachykinin-related peptide (TRP) signaling in cell culture. (A) To confirm the predicted membrane localization of the cloned *A. mellifera* TRP receptor (TRPR), Sf21 cells expressing the TRPR tagged with EGFP (green) were stained with the membrane plasma probe DiI (red) and nuclear probe Hoechst (blue). (B) Luciferase activity of HEK293 cells transfected with the reporter gene pCRE-Luc (CRE-L) and co-transfected with pFLAG-TRPR (TRPR) or vehicle vector (V) indicated that 1 μM treatment of TRP2 or TRP3 increases cAMP levels more than 10-fold. (C) Intracellular $Ca^2$ levels of HEK293 cells expressing TRPR rose sharply in response to TRP2 and TRP3, based on fluorescence measures of the $Ca^{2+}$ indicator Fura-2 AM. Hepes-buffered medium (HBM) was used as a control. (D) Dose-dependent response of ERK1/2 phosphorylation to TRP2 stimulation of Sf21 cells that expressed TRPR and were incubated with increasing doses of TRP2 (from 0.1 pM to 1 μM) before their harvest for Western blot analysis (log TRP2 (M) = logarithm of the molar concentration of TRP2).

The online version of this article includes the following source data and figure supplement(s) for figure 5:

**Source data 1.** Competitive binding assays with labeled tachykinin-related peptide 2 (TRP2) and TRP3.

**Source data 2.** Tachykinin-related peptide 2 (TRP2) and TRP3 exclusively activate its receptor (TRPR) and trigger cAMP signaling.

**Source data 3.** Tachykinin-related peptide 2 (TRP2)-induced cAMP accumulation is regulated by G-protein inhibitors and activators.

**Source data 4.** Intracellular $Ca^{2+}$ mobilization is induced by tachykinin-related peptide 2 (TRP2) and TRP3, and regulated by $G_{\alpha q}$ inhibitor.

**Source data 5.** Extracellular-signal-regulated kinase (ERK) phosphorylation levels in response to TRP/TRPR (tachykinin-related peptide and its receptor) signaling.

**Figure supplement 1.** Tachykinin-related peptide receptor (TRPR) localization and competitive binding of TRP2 and TRP3.

**Figure supplement 2.** cAMP generation is specific to tachykinin-related peptide 2 (TRP2) and TRP3 and dose-dependent TRP/TRPR (TRP and its receptor)-mediated cAMP accumulation in cells.

**Figure supplement 3.** TRP/TRPR (tachykinin-related peptide and its receptor) signaling induces cAMP accumulation via $G_{\alpha q}$ and $G_{\alpha s}$ pathways.

*Figure 5 continued on next page*

*Figure 5 continued*

**Figure supplement 4.** TRP/TRPR (tachykinin-related peptide and its receptor) signaling mediates intracellular $Ca^{2+}$ influx via $G_{\alpha q}$/PLC pathway.
**Figure supplement 5.** Extracellular-signal-regulated kinase (ERK) phosphorylation is dose- and time-dependent and can be inhibited.

signaling could be more widely conserved to adjust the context specificity of behavioral responses in animals.

Among all the signaling molecules in the nervous system, neuropeptides represent the largest and most diverse category and are crucial in orchestrating various biological processes and behavioral actions (*Burbach, 2011*; *Hökfelt et al., 2000*). Thus, we quantitatively compared the entire neuropeptidome among three behavioral worker phenotypes of AML and ACC without an a priori assumption. In addition to discovering several new neuropeptides from the ACC and AML brain, we identified TRP2 and TRP3 as candidates. TRPs have been associated with the modulation of appetitive olfactory sensation (*Ki et al., 2015*; *Winther et al., 2006*; *Gui et al., 2017*), foraging (*Pratavieira et al., 2018*), sex pheromone perception (*Shankar et al., 2015*), and aggression (*Asahina et al., 2014*). Particularly in honeybees, *TRP* is preferentially expressed in the mushroom body and some neurons scattered in the antennal and optic lobes (*Takeuchi et al., 2004*), and some expression has also been found in antennae (*Jain and Brockmann, 2020*). These expression patterns are consistent with our hypothesis that TRP signaling may be modulating behavioral responsiveness due to various inputs. Although specific knockdown of TRPs in *Drosophila* has revealed spatially variable effects (*Nässel et al., 2019*), the changes in response thresholds that we found are most likely due to systemic manipulation of TRP or TRPR levels. Global TRP expression in the honeybee worker brain also increases during the transition from nursing to foraging, further implicating it in the regulation of honeybee social behavior (*Pratavieira et al., 2014*; *Takeuchi et al., 2003*). The spatiotemporal expression of *TRPR* remains to be explored in *A. mellifera* and *A. cerana*, but the specific

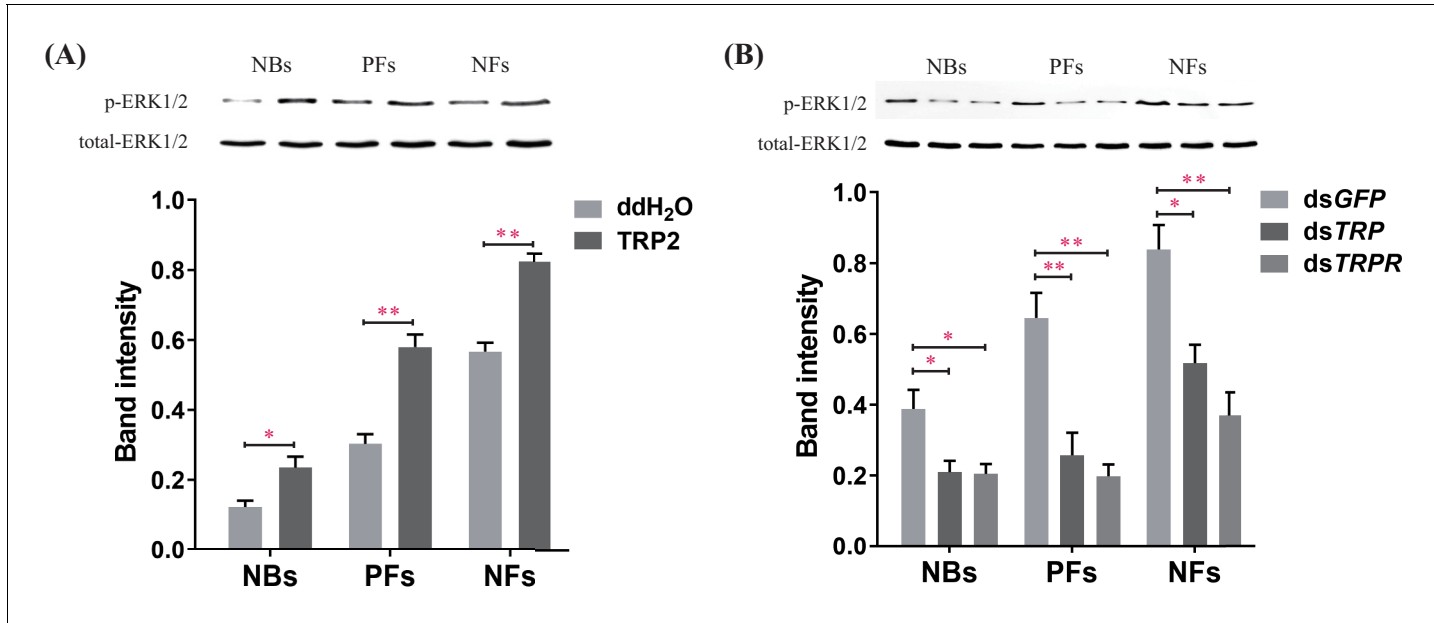

**Figure 6.** Manipulations of tachykinin-related peptide (TRP) and its receptor (TRPR) levels change extracellular-signal-regulated kinase (ERK) phosphorylation states in the brain of worker bees (*Apis mellifera ligustica*). (**A**) The ERK phosphorylation (p-ERK) levels after injection of TRP2 or ddH₂O into pollen foragers (PFs), nectar foragers (NFs), and nurse bees (NBs) of *A. mellifera ligustica*. (**B**) The p-ERK levels after transcript knockdown of *GFP*, *TRP*, or *TRPR* in PFs, NFs, and NBs. The p-ERK was normalized to a loading control (total-ERK). The data shown are representative of three independent experiments, and blots shown are representative of these experiments. Student's t-tests were used for pairwise comparisons between control and treatment groups within each behavioral phenotype (*p<0.05, **p<0.01, ***p<0.001).
The online version of this article includes the following source data for figure 6:

**Source data 1.** Extracellular-signal-regulated kinase (ERK) phosphorylation levels in response to manipulations of tachykinin-related peptide (TRP) and its receptor (TRPR) levels.

functions that have been linked to specific neurons in *Drosophila* (*Ki et al., 2015*) suggests that this information is critical for more detailed mechanistic models of the behavioral effects of tachykinin in honeybees.

In our study, only expression of TRP2 and TRP3 varied consistently among behavioral phenotypes of AML and ACC. In both species, TRP2 and TRP3 were most abundant in the brain of NFs, followed by PFs, and finally NBs. This is consistent with the very specific responsiveness of NBs to brood stimuli observed in our PER experiments, while the responsiveness of PFs and NFs was successively less specific: PFs responded specifically to two stimuli, while NFs did not show specifically strong responses to any stimuli. Moreover, the comparison between AML and ACC indicated higher TRP2 and TRP3 abundance in ACC in each behavioral phenotype, commensurate with the less specific PER responsiveness in ACC compared to AML. A few other neuropeptides, such as apidaecins, diuretic hormone, and prohormone-3, showed somewhat similar expression patterns in both species, but none of these was as tightly correlated to behavioral responsiveness and none has previously been connected with behavioral regulation in insects or other animals. Therefore, the TRPs were chosen as candidates of the control of honeybee division of labor for subsequent functional tests and molecular characterization.

The action of most insect neuropeptides is mediated by binding to GPCRs and often involves cAMP and $Ca^{2+}$ as second messengers (*Hauser et al., 2006*). The TRPR is activated by TRPs triggering intracellular cAMP accumulation and $Ca^{2+}$ mobilization in fruit flies and silkworms (*Bombyx mori*) (*He et al., 2014*; *Birse et al., 2006*), while no cAMP responses were discovered in stable flies (*Stomoxys calcitrans*) (*Poels et al., 2007*). The results of our peptide-based binding assays functionally confirmed that the honeybee TRPR is indeed the receptor for TRP2 and TRP3. The subsequent functional assays revealed that TRP signaling results in a dose-dependent increase in both intracellular cAMP and $Ca^{2+}$. Together, these results indicate that TRPs can activate TRPR and trigger second messengers to regulate downstream functions. TRP2 displayed a higher affinity to TRPR and induced higher cAMP and $Ca^{2+}$ signaling than TRP3, leading us to focus on TRP2 in the later in vivo experiments. Moreover, TRP signaling is sensitive to $G_{\alpha s}$ activation and is significantly blocked by $G_{\alpha q}$ and PKA inhibitors, suggesting both $G_{\alpha s}$ and $G_{\alpha q}$ are involved in TRP signaling in honeybees. Many GPCRs are able to induce mitogen-activated protein kinase cascades via cooperation of $G_{\alpha s}$, $G_{\alpha q}$, and $G_{\alpha i}$ signals, leading to the phosphorylation of ERK1/2, which plays critical roles in diverse biological processes (*Rozengurt, 2007*). Our results indicate that honeybee TRP signaling mediates phosphorylation of ERK1/2 in a dose- and time-dependent manner in both HEK293 and Sf21 cells. In addition, ERK1/2 activation was significantly inhibited by the PKA, PKC, and MEK inhibitors, which is in line with the observation of intracellular cAMP accumulation and $Ca^{2+}$ mobilization. Thus, honeybees seem to be very similar to silkworms with regard to the involvement of the $G_{\alpha s}$/cAMP/PKA and $G_{\alpha q}$/$Ca^{2+}$/PKC signaling pathways in the regulation of TRP-induced ERK1/2 activation (*He et al., 2014*). Taken together, our results demonstrate that the honeybee TRPR is specifically activated by TRPs, eliciting intracellular cAMP accumulation, $Ca^{2+}$ mobilization, and ERK phosphorylation by dually coupling $G_{\alpha s}$ and $G_{\alpha q}$ signaling pathways.

Our in vitro and in vivo demonstrations that TRP signaling activates the ERK pathway indicated that the basic signaling mechanisms are conforming to the general patterns in insects (*Hauser et al., 2006*; *He et al., 2014*; *Werry et al., 2005*). Thus, our behavioral results may extend to other species. ERK links TRP signaling also to the insulin/insulin-like signaling (IIS) pathway. IIS is controlled by neuropeptides through ERK in *Drosophila* (*Lee et al., 2008*), and this connection in honeybees ties TRP back to the age-based division of labor among workers: IIS influences the timing of the behavioral maturation of honeybee workers and brain *AmIlp1* is significantly higher expressed in foragers than nurses (*Ament et al., 2008*), consistent with our finding that TRPs are higher in foragers than nurses. Numerous other physiological changes accompany the transition from in-hive nurses to foragers (*Scheiner et al., 2006*; *Robinson, 1987*; *Wang et al., 2012*; *Toth and Robinson, 2005*, and our results integrate TRPs as the most important neuropeptides into the regulation of the behavioral ontogeny of honeybee workers and potential feedback loops to the modulation of behavioral response thresholds. The specialization of NFs and PFs has also been linked to IIS (*Hunt et al., 2007*; *Wang et al., 2009*) and explained by differences in sucrose response thresholds (*Pankiw and Page, 2000*). Our findings here may connect the differences in response thresholds and IIS mechanistically through the TRP and ERK signaling pathways.

The PER paradigm is well suited to test behavioral response thresholds and has been used for over 50 years in honeybees (*Giurfa and Sandoz, 2012*). Consistent with previous studies, we found PFs to be more responsive to sucrose than NFs and NBs in *A. mellifera* (*Page et al., 1998*). Moreover, we found corresponding differences between these behavioral groups in the closely related *A. cerana*. The PF's responsiveness to low sucrose concentrations might also make them more responsive to pollen, but the causation of the PER to pollen is unclear (*Grüter et al., 2008*) and other components of pollen may functionally distinguish pollen from sucrose responsiveness (*Nicholls and de Ibarra, 2013*). Our results support the view that pollen and sucrose are distinct stimuli: While our experimental manipulations of TRP signaling altered the responsiveness of PFs to pollen and sucrose, only responsiveness to sucrose was affected in NFs and only responsiveness to larvae was affected in NBs. The functional significance of the PER in response to larvae is currently unclear, but we show that it is specific to nurses and it has previously been linked to brood provisioning (*Zhang et al., 2020*). Thus, our diverse PER results in two species comprehensively support the hypothesis that task-specific response thresholds guide behavioral specialization, leading to division of labor among honeybee workers (*Theraulaz et al., 1998*; *Beshers et al., 1999*; *Robinson, 1992*).

TRPs may adjust specific sensory neural circuits, potentially acting in concert with other neuromodulators (*Kahsai et al., 2010*; *Jung et al., 2013*). However, we have currently no evidence to support the hypothesis of different molecular TRP actions in different stimulus-response pathways (cf *Ki et al., 2015*). Thus, we have to restrict our conclusion to the simple hypothesis that TRP signaling

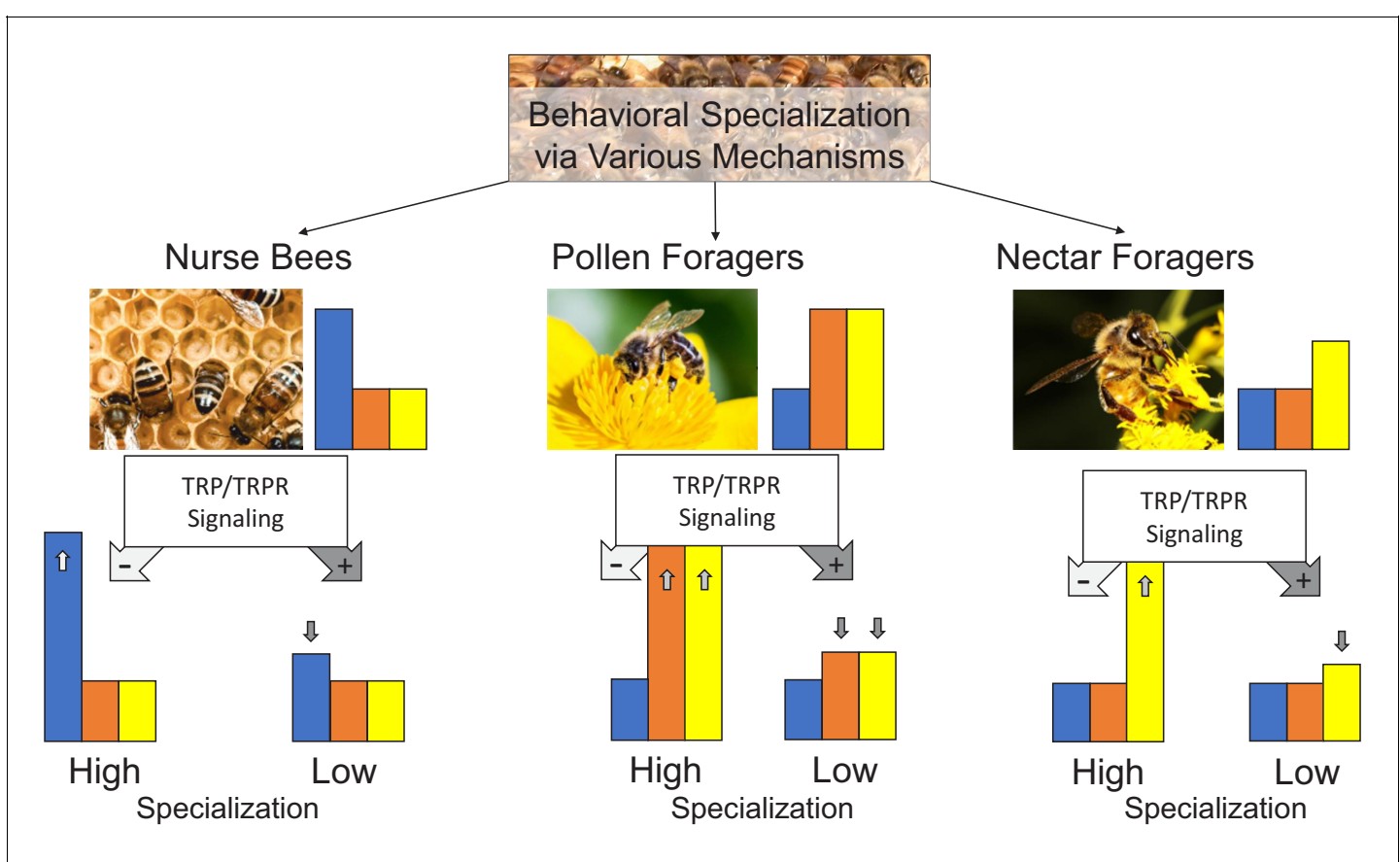

**Figure 7.** Conceptual model representation the TRP signaling effect. Worker honeybees specialize into nurse bees, pollen foragers, and nectar foragers through various influences and display task-specific response thresholds to brood (blue bar), pollen (orange bar), and nectar (yellow bar) stimuli. The tachykinin pathway (tachykinin-related peptide [TRP] and its receptor [TRPR] signaling) can regulate the extent of specialization by specifically decreasing (pathway active: dark gray arrow with + symbol) or increasing (pathway inactive: light gray arrow with – symbol) the responsiveness of each behavioral phenotype to its specific stimuli. This study demonstrates the task-specific response thresholds and how TRP/TRPR signaling affects these thresholds. The predicted behavioral effects of increasing (high) or decreasing (low) the specialization of the behavioral phenotypes remains to be demonstrated.

decreases task-specific response thresholds of behavioral specialists without affecting non-task-related thresholds: It decreases pollen and sucrose responsiveness in PFs, sucrose responsiveness in NFs, and responsiveness to larvae in NBs. TRP signaling in honeybees may thus be a general regulator of how task-specific stimuli are weighted relative to others and consequently how specialized behavioral specialists are (*Figure 7*). This effect translates to different degrees of division of labor in social insect colonies and may control the context specificity of behavioral responses in animals more generally (*Sih et al., 2004*).

Although AML and ACC are close relatives with similar basic biology, some behavioral differences have evolved since their speciation (*Oldroyd and Wongsiri, 2009*). AML and ACC share the age-based division of labor, with younger bees specializing on nursing before maturing to foraging activities (*Hepburn and Radloff, 2011*) and ACC foragers also specialize in nectar or pollen collection (*Ji et al., 2020*; *Rueppell et al., 2008*) similar to AML (*Page et al., 2000*). Accordingly, we found the main differences of stimulus responsiveness and TRPs' expression among worker phenotypes conserved. However, ACC exhibited less responses to the task-specific stimuli than AML. Consistent PER differences in AML and ACC between NFs and PFs and a generally lower responsiveness of ACC have been identified before (*Yang et al., 2013*), but the biological interpretation has remained unclear. It is possible that the species differences arise due to methodological bias, favoring AML performance in PER assays. However, our study offers the alternative explanation that ACC workers are less specialized than AML workers due to higher TRP signaling. Lower innate specialization may accompany better learning of ACC (*Qin et al., 2012*), facilitating its more opportunistic worker task allocation and resource exploitation than AML (*Tan et al., 2008*). These alternative life history strategies are plausible, given the typical differences in colony size and habitat (*Jung et al., 2013*; *Sih et al., 2004*; *Ruttner, 1988*). All three worker phenotypes of ACC exhibited higher levels of TRPs than their AML counterparts, but functional verification at the level of colony phenotypes will be required to unambiguously link TRP signaling to such interspecific differences in life history.

## Materials and methods

### Key resources table

| Reagent type (species) or resource | Designation | Source or reference | Identifiers | Additional information |
|---|---|---|---|---|
| Gene (*Apis mellifera*) | *TRP* | BEEBASE | GB49248 | |
| Gene (*Apis mellifera*) | *TRPR* | BEEBASE | GB49973 | |
| Strain, strain background (*Escherichia coli*) | DH5α competent cells | TaKaRa | Cat#9057 | |
| Cell line (*Homo sapiens*) | HEK293 (embryonic kidney cell line) | ATCC | Cat#CRL-1573, RRID:CVCL_0045 | |
| Cell line (*Spodoptera frugiperda*) | Sf21 (pupal ovary cell line) | Thermo Fisher Scientific | Cat#11497013, RRID:CVCL_0518 | |
| Antibody | Anti-ERK1/2 (rabbit monoclonal) | Cell Signaling Technology | Cat#4695T; RRID:AB_390779 | (1:1000) |
| Antibody | Anti-phospho-ERK1/2 (rabbit polyclonal ) | Cell Signaling Technology | Cat#9101S; RRID:AB_331646 | (1:1000) |
| Antibody | Anti-rabbit IgG, HRP-linked (goat, polyclonal) | Cell Signaling Technology | Cat#7074S; RRID:AB_2099233 | (1:5000) |
| Recombinant DNA reagent | pCMV-FLAG (plasmid) | Sigma-Aldrich | Cat#E 8770 | |
| Recombinant DNA reagent | pEGFP-N1 (plasmid) | Clontech | Cat#6085–1 | |
| Sequence-based reagent | *TRPRf_F* | This paper | PCR primers | See *Supplementary file 6* |
| Sequenced-based reagent | *TRPRf_R* | This paper | PCR primers | See *Supplementary file 6* |

*Continued on next page*

*Continued*

| Reagent type (species) or resource | Designation | Source or reference | Identifiers | Additional information |
|---|---|---|---|---|
| Sequence-based reagent | *TRPRe_F* | This paper | PCR primers | See *Supplementary file 6* |
| Sequence-based reagent | *TRPRe_R* | This paper | PCR primers | See *Supplementary file 6* |
| Sequence-based reagent | *TRPRi_F* | This paper | RNAi primers | See *Supplementary file 6* |
| Sequence-based reagent | *TRPRi_R* | This paper | RNAi primers | See *Supplementary file 6* |
| Sequence-based reagent | *TRPRq_F* | This paper | qPCR primers | See *Supplementary file 6* |
| Sequence-based reagent | *TRPRq_R* | This paper | qPCR primers | See *Supplementary file 6* |
| Sequence-based reagent | *TRPi_F* | This paper | RNAi primers | See *Supplementary file 6* |
| Sequence-based reagent | *TRPi_R* | This paper | RNAi primers | See *Supplementary file 6* |
| Sequence-based reagent | *TRPq_F* | This paper | qPCR primers | See *Supplementary file 6* |
| Sequence-based reagent | *GFPi_F* | This paper | RNAi primers | See *Supplementary file 6* |
| Sequence-based reagent | *GFPi_R* | This paper | RNAi primers | See *Supplementary file 6* |
| Sequence-based reagent | *TRPq_R* | This paper | qPCR primers | See *Supplementary file 6* |
| Commercial assay or kit | PrimeScript RT reagent kit | TaKaRa | Cat#RR047A | |
| Commercial assay or kit | TB Green Fast qPCR Mix | TaKaRa | Cat#RR430A | |
| Commercial assay or kit | Luciferase assay system | Promega | Cat#E1500 | |
| commercial assay or kit | T7 RiboMAX Express RNAi System | Promega | Cat#P1700 | |
| Chemical compound, drug | TAMRA-ALMGFQGVRa | SynPeptide | | |
| Chemical compound, drug | TAMRA-APMGFQGMRa | SynPeptide | | |
| Chemical compound, drug | ALMGFQGVR | SynPeptide | | |
| Chemical compound, drug | APMGFQGMRa | SynPeptide | | |
| Chemical compound, drug | CBR-5884 | Sigma-Aldrich | SML1656 | |
| Chemical compound, drug | Pertussis toxin | Tocris Bioscience | Cat#3097/50U | CAS: 70323-44-3 |
| Chemical compound, drug | H89 | Tocris Bioscience | Cat#2910/1 | CAS: 130964-39-5 |
| Chemical compound, drug | U73122 | Tocris Bioscience | Cat#1268/10 | CAS: 112648-68-7 |
| Chemical compound, drug | Cholera toxin | Tocris Bioscience | Cat#HY-P1446 | CAS: 9012-63-9 |
| Chemical compound, drug | YM-254890 | Tocris Bioscience | Cat#HY-111557 | CAS: 568580-02-9 |
| Software, algorithm | SPSS Statistics 20.0 | IBM | RRID:SCR_019096 | |
| Software, algorithm | Xcalibur 3.0 | Thermo Fisher Scientific | RRID:SCR_014593 | |
| Software, algorithm | PEAKS 8.5 | Bioinformatics Solutions | | |
| Software, algorithm | Gene cluster 3.0 | *de Hoon et al., 2004* | https://doi.org/10.1093/bioinformatics/bth078 | |

*Continued*

| Reagent type (species) or resource | Designation | Source or reference | Identifiers | Additional information |
|---|---|---|---|---|
| Software, algorithm | Primer Premier 5.0 | PREMIER Biosoft | | |
| Other | Hoechst 33342 stain | Beyotime | Cat#C1027 | (1 µg/ml) |
| Other | DiI stain | Beyotime | Cat#C1036 | (1 µg/ml) |
| Other | Lipo6000 transfection reagent | Beyotime | Cat#C0526 | |
| Other | LipoInsect transfection reagent | Beyotime | Cat#C0551 | |

## Honeybee sources and sampling

Two honeybee species, AML and ACC, were maintained in the apiary of the Institute of Apicultural Research at the Chinese Academy of Agricultural Sciences in Beijing. Three colonies of each species with mated queens of identical age were selected as experimental colonies, and before experiments the colonies were equalized in terms of adult bee population, brood combs, and food storage. Frames containing old pupae (1–2 days before emergence) were put into an incubator (34℃ and 80% relative humidity) for eclosion. Newly emerged worker bees were paint-marked on their thoraxes and placed back into their parent colonies. Ten days later, marked bees that had their head and thorax in open brood cells while contracting their abdomen for more than 10 s were collected as NBs. Twenty days after eclosion, marked bees were collected during early morning (between 8:00 a.m. and 10:00 a.m.) in good weather conditions during the blooming period of black locusts (*Robinia pseudoacacia* L.) as forager bees. The entrance to the hives were blocked to facilitate collecting. Bees flying into the hive with pollen loads were collected as PFs, returning foragers without pollen loads were collected as NFs. The experimental design of six groups (three behavioral phenotypes in two species) was used to compare responsiveness to task-specific stimuli ('Comparative PER experiments' section) and to relate these phenotypes to differences in the brain neuropeptidome ('Quantitative comparisons of brain neuropeptidomes' section).

## Comparative PER experiments

To investigate the responsiveness of different worker bee behavioral phenotypes (NBs, PFs, and NFs of AML and ACC) to different stimulus modalities (sucrose solution, pollen, and larva), series of PER experiments were performed. One hundred bees of each behavioral phenotype were collected from each experimental colony in the morning, transferred to the laboratory and narcotized on ice, then harnessed using a previously described protocol (*Wang and Tan, 2014*). All harnessed bees were fed to satiation with 50% sucrose solution and placed in a dark incubator (20℃ and 65% relative humidity) overnight. After 24 hr, all surviving bees were assayed for their PER following the methodology of *Page et al., 1998*. Each stimulus was assessed independently with a new set of bees.

To investigate the sucrose responsiveness, bees were assayed using an ascending order of sucrose concentrations: 0.1%, 0.3%, 1%, 3%, 10%, and 30% (weight/weight). A small droplet of each solution was touched to the bees' antennae for 3 s and the extension of the proboscis was monitored during this time. The interval between each sucrose solution trial was 5 min to exclude sensitization or habituation effects. The total number of PER responses after stimulation with the six different sucrose concentrations was combined into an SRS of a bee (*Pankiw et al., 2001*; *Scheiner et al., 2002*; *Scheiner et al., 2003*). The SRSs of the three behavioral phenotypes in the same species were compared using Kruskal-Wallis tests with Bonferroni correction. Pairwise Mann-Whitney U tests were used for comparing the same phenotype from two honeybee species. The sucrose responsiveness for specific sucrose concentrations was further compared between different groups with independent chi-square tests.

To test pollen stimulation, fresh pollen loads that had been removed from the leg of randomly selected PFs of the test group were used: AML were tested with pollen collected by AML foragers and ACC with pollen collected by ACC foragers. These loads contained a mixture of different pollen, predominated by black locust (*R. pseudoacacia*). As a control for mechanical stimulation, each bee

had both antennae first touched with a piece of filter paper and spontaneous responders were excluded. Subsequently, both antennae of each bee were gently touched with a pollen load and PER responses were recorded. The pollen responsiveness was compared with independent chi-square tests between different groups.

To test responsiveness to larva, 1-day-old larvae from each honeybee species were collected, briefly rinsed in distilled water to remove royal jelly residue and dried on a filter paper. As before, both antennae of bees were touched with a piece of filter paper first and spontaneous responders were excluded, then PERs in response to a larva touching the antennae were recorded. The responsiveness to larvae was compared with independent chi-square tests between different groups. Statistical analyses were conducted using SPSS Statistics 20.0 (IBM, USA).

## Quantitative comparisons of brain neuropeptidomes

To explore brain neuropeptide functions in behavioral regulation, a label-free quantitative strategy was employed to compare neuropeptidomic variations between behavioral phenotypes and the two honeybee species. Three independent biological replicate samples (120 bees per sample) of NBs, PFs, and NFs of both AML and ACC (18 samples total) were collected and immediately frozen in liquid nitrogen. Individual brains were carefully dissected from the head capsule while remaining chilled on ice, and the dissected brains were frozen at $-80°C$ until neuropeptide extraction.

The brains were homogenized at $4°C$ by using a 90:9:1 solution of methanol, $H_2O$, and acetic acid. The homogenates were centrifuged at 12,000 g for 10 min at $4°C$. The resulting supernatant containing the neuropeptides was collected and dried. The extracted neuropeptide samples were dissolved in 0.1% formic acid in distilled water, and the peptide concentration was quantified using a Nanodrop 2000 spectrophotometer (Thermo Fisher Scientific, USA). LC-MS/MS analysis was performed on an Easy-nLC 1200 (Thermo Fisher Scientific) coupled Q-Exactive HF mass spectrometer (Thermo Fisher Scientific). Buffer A (0.1% formic acid in water) and buffer B (0.1% formic acid in acetonitrile) were used as mobile phase buffers. Neuropeptides were separated using the following gradients: from 3% to 8% buffer B in 5 min, from 8% to 20% buffer B in 80 min, from 20% to 30% buffer B in 20 min, from 30% to 90% buffer B in 5 min, and remaining at 90% buffer B for 10 min. The eluted neuropeptides were injected into the mass spectrometer via a nano-ESI source (Thermo Fisher Scientific). Ion signals were collected in a data-dependent mode and run with the following settings: full scan resolution at 70,000, automatic gain control (AGC) target $3 \times 10^6$; maximum inject time (MIT) 20 ms; scan range m/z 300–1800; MS/MS scans resolution at 17,500; AGC target $1 \times 10^5$; MIT 60 ms; isolation window 2 m/z; normalized collision energy 27; loop count 10; and dynamic exclusion:charge exclusion: unassigned, 1, 8, >8; peptide match preferred; exclude isotopes on; dynamic exclusion: 30 s. Raw data were retrieved using Xcalibur 3.0 software (Thermo Fisher Scientific).

The extracted MS/MS spectra were searched against a composite database of *A. mellifera* (23,491 protein sequences, downloaded from NCBI on July 2018) or *A. cerana* (20,934 protein sequences, downloaded from NCBI on July 2018) using in-house PEAKS 8.5 software (Bioinformatics Solutions, Canada). Amidation (A, $-0.98$) and pyro-glu from Q (P, $-17.03$) were selected as variable modifications. The other parameters used were: parent ion mass tolerance, 20.0 ppm; fragment ion mass tolerance, 0.05 Da; enzyme, none; maximum allowed variable PTM per peptide, 2. A fusion target-decoy approach was used for the estimation of the false discovery rate and controlled at $\leq 1.0\%$ ($-10$ log p$\geq 20.0$) both at protein and peptide levels. Neuropeptide identifications were only used if $\geq 2$ spectra were identified in at least two of the three replicates of each sample type.

Quantitative comparison of brain neuropeptidomes was performed by the label-free approach in PEAKS Q module. Feature detection was performed separately on each sample by using the expectation-maximization algorithm. The features of the same peptide from different samples were reliably aligned together using a high-performance retention time alignment algorithm (*Lin et al., 2013*). Peptide features were considered significantly different between experimental groups if pairwise p<0.01 and fold change $\geq 1.5$. A heat map of differentially expressed proteins was created by Gene cluster 3.0 using the unsupervised hierarchical clustering, and the result was visualized using Java Tree view software. The LC−MS/MS data and search results are deposited in ProteomeXchange Consortium (http://proteomecentral.proteomexchange.org) via the PRIDE partner repository with the dataset identifier PXD018713.

## Characterization of honeybee TRP signaling pathway

To characterize honeybee TRP signaling pathway, the TRPR gene was first cloned and expressed in human and insect cell lines to identify its cellular location and verify its binding to TRPs. Additionally, these cells were used to test whether TRP/TRPR signaling triggers intracellular cAMP accumulation, $Ca^{2+}$ mobilization, and ERK phosphorylation.

### TRPR gene clone and expression

To amplify the full-length sequence encoding *TRPR* of *A. mellifera*, primers were designed using Primer Premier 5.0 software (PREMIER Biosoft, USA) based on the sequence from GenBank KT232312. The coding sequence of TRPR was amplified and cloned into FLAG-tag expression vectors (pCMV-FLAG and pBmIE1-FLAG) and EGFP-tag expression vectors (pEGFP-N1 and pBmIE1-EGFP). The primers used are documented in *Supplementary file 6*. All constructs were sequenced to verify the correct sequence, orientation, and reading frame of the inserts.

The human embryonic kidney cell line HEK293 and the insect *S. frugiperda* pupal ovary cell line Sf21 were used for honeybee TRPR expression. HEK293 cells (RRID:CVCL_0045) were purchased from American Type Culture Collection (ATCC, CRL-1573, the identity has been authenticated using STR profiling) and cultured in DMEM medium (Gibco, USA) supplemented with 10% fetal bovine serum (FBS). Cells were routinely tested for mycoplasma contamination every 6 months. Sf21 cells (RRID: CVCL_0518) were purchased from Thermo Fisher Scientific and were cultured in TC100 medium (Gibco) supplemented with heat-inactivated 10% FBS. Cells were routinely tested for mycoplasma contamination every 6 months. Transfection of HEK293 cells was performed using Lipo6000 transfection reagent (Beyotime, China), while transfection of Sf21 cells was performed using LipoInsect transfection reagent (Beyotime), according to the manufacturer's instructions.

### Cellular location of TRPR

To confirm the location of the honeybee TRPR, receptor surface expression assays were performed. HEK293 or Sf21 cells expressing TRPR-EGFP were seeded onto poly-L-lysine-coated glass coverslips and allowed to attach overnight under normal growth conditions. After 24 hr, cells were incubated with the membrane probe DiI (Beyotime) and the nucleic acid probe Hoechst 33342 (Beyotime) at 37℃ for 10 min, then fixed with 4% paraformaldehyde for 15 min. Cells transfected with empty EGFP-tag expression vectors were used as a control. The cells were imaged using a Leica SP8 (Leica Microsystems, Germany) confocal microscope equipped with an HC PL APO CS2 63×/1.40 oil objective. Images were acquired with the sequence program in the Leica LAS X software.

### Binding of TRPs to TRPR

To confirm the direct binding of the honeybee TRPs to TRPR, competitive binding experiments were performed using synthesized TAMRA-TRP2 (TAMRA-ALMGFQGVRa) and TAMRA-TRP3 (TAMRA-APMGFQGMRa), with TAMRA labeled at the N-terminus. The neuropeptides used as ligands here and in later sections were commercially synthesized by SynPeptide Co, Ltd (China). All peptides were purified by reverse-phase high-performance liquid chromatography with a purity >98%, lyophilized, and diluted to the desired concentrations for subsequent experiments. The peptide sequences were verified by us using a Q-Exactive HF mass spectrometer (Thermo Fisher Scientific).

HEK293 and Sf21 cells expressing FLAG-TRPR were first seeded onto poly-L-lysine-coated 96-well plates and cultured overnight. On the next day, cells were washed once with phosphate-buffered saline (PBS), then incubated with 25 ml TAMRA-TRP2 or TAMRA-TRP3 (10 nM) in the presence of increasing concentrations of unlabeled TRP2 and TRP3 in a final volume of 100 ml of binding buffer (PBS containing 0.2% bovine serum albumin). Cells were incubated at room temperature for 2 hr. Fluorescence intensity was measured with a fluorescence spectrometer microplate reader (Tecan Infinite 200 PRO, Tecan, Switzerland) after washing twice with binding buffer. The cells transfected with empty FLAG-tag expression vectors were used as a control. The binding displacement curves were analyzed by GraphPad Prism 8.0 (GraphPad Software, USA) using the non-linear logistic regression method.

## TRP/TRPR signaling targets: cAMP, Ca$^{2+}$, and ERK

To test whether TRP/TRPR signaling affects cAMP accumulation, intracellular cAMP was measured after incubation of HEK293 and Sf21 cells expressing FLAG-TRPR and pCRE-Luc with TRP2 and TRP3. After seeding in a 96-well plate overnight, HEK293 or Sf21 cells co-transfected with pFLAG-TRPR and pCRE-Luc were grown to about 90% confluence. After washing once with PBS, cells were incubated with the neuropeptides TRP2, TRP3, SNF, PDH, and CRZ in serum-free medium for 4 hr at 37°C for HEK293 cells, and at 28°C for Sf21 cells. Cells transfected with empty EGFP-tag expression vectors were used as a control. Luciferase activity was detected by a luciferase assay system (Promega, USA). Fluorescence intensity was measured with a Tecan fluorescence spectrometer. When characterizing the TRP-mediated cAMP accumulation, cells were pretreated with G$_{\alpha i}$ inhibitor PTX, G$_{\alpha s}$ activator cholera toxin (CTX), G$_{\alpha q}$ inhibitor YM-254890, and PKA inhibitor H89 before stimulation with TRP2.

To test whether TRP signaling also affects intracellular Ca$^{2+}$ concentrations, intracellular Ca$^{2+}$ was measured after incubation of HEK293 and Sf21 cells expressing FLAG-TRPR with TRP2 or TRP3. Cells were detached by a non-enzymatic cell dissociation solution (Sigma-Aldrich, USA), washed twice with PBS, and resuspended at a density of $5 \times 10^6$ cells/ml in HEPES buffered saline (Macklin, China). Cells were then incubated with 3 µM Fura-2 AM (MedChemExpress, USA) for 30 min at 37°C for HEK293 cells, and at 28°C for Sf21 cells. Intracellular Ca$^{2+}$ flux was measured using excitation wavelengths alternating at 340 and 380 nm with emission measured at 510 nm in a Tecan fluorescence spectrometer. When characterizing the detailed TRP-mediated intracellular Ca$^{2+}$ mobilization, cells were pretreated with G$_{\alpha q}$ inhibitor YM-254890 and PLC inhibitor U73122 before stimulation with TRP2.

To assess whether TRP signaling mediates ERK1/2 signaling, ERK1/2 phosphorylation was measured by Western blot analysis after incubation of HEK293 and Sf21 cells expressing FLAG-TRPR with TRP2. Cells were seeded in 24-well plates and starved for 4 hr in serum-free medium to reduce background ERK1/2 activation and eliminate the effects of the change of medium. After incubation with TRP2, cells were lysed by RIPA buffer (Beyotime) at 4°C for 30 min. Protein concentration was determined according to the Bradford method using BSA as the standard and the absorption was measured at 595 nm (spectrophotometer DU800, Beckman Coulter, USA), then all the samples were kept in −80°C for further use. For Western blot, equal amounts of total cell lysate (20 µg/lane) were fractionated by SDS-PAGE (10%) and transferred to a PVDF membrane (Millipore, USA) using an iBlot dry blotting system (Invitrogen, USA). The membranes were blocked for 2 hr at room temperature and then incubated with rabbit monoclonal anti-pERK1/2 antibody (Cell Signaling Technology, USA) and anti-rabbit horseradish peroxidase-conjugated secondary antibody (Cell Signaling Technology) according to the manufacturers' protocols. Antibody reactive bands were visualized using Pierce ECL western blotting substrate (Thermo Fisher Scientific, USA) followed by photographic film exposure. Total ERK1/2 was assessed as a loading control after p-ERK1/2 chemiluminescence detection. Quantification analyses were performed using Gel-Pro Analyzer 4.0 software (Media Cybernetics, USA).

To explore the detailed TRP-mediated ERK1/2 signaling, cells were pretreated with G$_{\alpha i}$ inhibitor PTX, MEK inhibitor U0126, PKA inhibitor H89, and PKC inhibitor Go6983 before stimulation with TRP2.

## Effects of TRP2 injection on task-specific responsiveness

To confirm the function of TRP on task-specific responsiveness, NBs, PFs, and NFs of AML were injected with TRP2 and tested for their PER response to sucrose solution, pollen, and larva. About 150 bees of each behavioral phenotype were collected in the morning, then harnessed, fed and placed in a dark incubator as described in 'Comparative PER experiments' section. After 24 hr, all surviving bees were evenly divided into two groups and injected with 1 µl TRP2 solution (1 µg/µl, synthesized TRP2 dissolved in ddH$_2$O) or 1 µl of ddH$_2$O into the head of honeybees via the central ocellus using a glass capillary needle coupled to a microinjector. Bees injected with ddH$_2$O were used as control. All injected bees were put back to the dark incubator and 1 hr after injection all surviving bees were assayed for their PER to stimulations of sucrose solution, pollen, and larva as described in 'Comparative PER experiments' section. Each experiment was performed with a new set of bees containing about 55 individuals per experimental and control group.

The average SRSs of the TRP2 injection group and the ddH$_2$O injection group were compared separately for each of the three behavioral phenotypes (NBs, PFs, and NFs) using pairwise Mann-Whitney U tests. The sucrose responsiveness was further compared between different groups at each specific sucrose concentration with independent chi-square tests. The responsiveness to pollen and larvae was compared between TRP2 injection group and ddH$_2$O injection group with independent chi-square tests for each behavioral phenotype separately. All statistical analyses were performed with SPSS Statistics 20.0 (IBM).

## Effects of RNAi-mediated downregulation of *TRP* or *TRPR* on responsiveness

To further confirm the hypothesized effects of TRP/TRPR signaling on task-specific responsiveness, RNAi-mediated downregulation of *TRP* and *TRPR* was performed on NBs, PFs, and NFs of AML and then their PER to sucrose solution, pollen, and larva was compared to controls.

Before evaluating the behavioral effects of transcript knockdown of *TRP* or *TRPR*, preliminary experiments were performed to test the dsRNA-mediated knockdown efficiencies of *TRP* and *TRPR*. The dsRNAs of the *TRP* and *TRPR* genes were prepared using the T7 RiboMAX Express RNAi system (Promega). The primers used are listed in *Supplementary file 6*. Sixty bees were randomly collected from each of the three AML colonies. Bees were harnessed, fed with sucrose, and put into the dark incubator (20℃ and 65% relative humidity) to acclimatize to the experimental conditions. After 30 min, dsRNA (200 ng/bee for *TRP*, 2 μg/bee for *TRPR*) was microinjected into the head of honeybees via the central ocellus using a glass capillary needle-coupled microinjector. dsRNA of green fluorescent protein gene (ds*GFP*, 2 μg/bee) was used as control in all RNAi experiments. All harnessed bees were fed with 50% sucrose solution every 12 hr. At 0, 12, 24, and 48 hr after injection, a group of six individual bees were collected from each injection group (ds*TRP*, ds*TRPR*, and ds*GFP*) for comparing *TRP* and *TRPR* expression. Individual brains were carefully dissected and frozen at −80℃ until RNA extraction. Three independent replicate groups per condition were collected and qRT-PCR was performed to calculate the RNAi efficiency. Total RNA was isolated using TRIzol reagent (Takara, Japan). Total RNA quantification was performed by NanoDrop 2000 spectrophotometer (Thermo Fisher Scientific), and the quality of RNA was evaluated by 1.0% denaturing agarose gel electrophoresis. Reverse transcription was performed using a PrimeScript RT reagent kit (Takara), according to the manufacturer's instructions. Gene-specific mRNA levels were assessed by qPCR using TB Green Fast qPCR Mix (Takara) on a LightCycler 480II instrument (Roche, Switzerland). The *β-actin* gene was used as a reference gene. After verifying amplification efficiency of the selected genes and β-actin (from 96.8% to 100.5%), the differences in gene expression levels were calculated using the $2^{-\Delta\Delta Ct}$ method. Pairwise differences in gene expression were considered significant at p<0.05, using one-way ANOVA (SPSS Statistics 20.0). The primers used for qPCR are shown in *Supplementary file 6*.

After determination of knockdown efficiencies (see 'Results' section), 24 hr post-injection was chosen as the time point to study the PER effects of dsRNA-mediated knockdown of *TRP* and *TRPR*. About 200 bees of each behavioral phenotype (NBs, PFs, and NFs of AML) were collected in the morning, harnessed, and remained in a dark incubator to acclimatize. After 30 min, all surviving bees of each behavioral phenotype were evenly divided into three groups, injected with ds*TRP*, ds*TRPR*, and ds*GFP*, and kept as described above. After 24 hr, all surviving bees were assayed for their PER to stimulations of sucrose solution, pollen, or larva as described in 'Comparative PER experiments' section. Each stimulus was assessed with a new set of bees containing about 55 individuals for each treatment group (ds*TRP*, ds*TRPR*, and ds*GFP*). The SRSs of the *TRP*-knockdown, *TRPR*-knockdown, and control groups were compared using Kruskal-Wallis tests with Bonferroni correction for each behavioral phenotype separately. The sucrose responsiveness was further compared between the different groups at the same sucrose concentration with independent chi-square tests. The responsiveness to pollen and larvae was compared between the *TRP* knockdown, *TRPR* knockdown, and control groups using independent chi-square tests for each behavioral phenotype separately. All statistical analyses were performed with SPSS Statistics 20.0 (IBM).

## Effects of TRP2 injection and RNAi-mediated downregulation of *TRP* and *TRPR* on ERK signaling in honeybee workers

To test whether manipulating TRP/TRPR signaling has effect on honeybee ERK signaling, a group of 10 individual worker bees were collected from each injection group (ddH$_2$O, TRP2, ds*TRP*, ds*TRPR*, and ds*GFP*) to compare ERK phosphorylation levels. Three independent replicate groups per condition were collected and Western blot analyses were performed: Honeybee brains were carefully dissected and frozen at −80℃ until protein extraction. Brain protein extractions were carried out according to our previously described method with some modifications. Briefly, the larvae were homogenized with lysis buffer (LB, 8 M urea, 2 M thiourea, 4% CHAPS, 20 mM Tris-base, 30 mM dithiothreitol). The mixture was homogenized for 30 min on ice and sonicated 20 s per 5 min during this time, then centrifuged at 12,000 g and 4℃ for 10 min. Ice-cold acetone was added to the collected supernatants, and then the mixture was kept on ice for 30 min for protein precipitation. Subsequently, the mixture was centrifuged at 12,000 g and 4℃ for 10 min. The supernatant was discarded and the pellets were resolved in LB and kept at −20℃ for further use. Western blot analyses were performed as described in 'TRP/TRPR signaling targets: cAMP, Ca$^{2+}$, and ERK' section.

## Acknowledgements

We appreciate Dr. Huipeng Yang at Institute of Apicultural Research for kind gifts of vectors.

## Additional information

### Funding

| Funder | Grant reference number | Author |
| --- | --- | --- |
| National Natural Science Foundation of China | 31970428 | Bin Han |
| Agricultural Science and Technology Innovation Program | CAAS-ASTIP-2015-IAR | Jianke Li |
| National Project for Upgrading the Beekeeping Industry of China | | Jianke Li |
| Earmarked Fund for Modern Agro-industry Technology Research System | CARS-44 | Jianke Li |
| University of North Carolina at Greensboro | | Olav Rueppell |

The funders had no role in study design, data collection and interpretation, or the decision to submit the work for publication.

### Author contributions

Bin Han, Conceptualization, Data curation, Formal analysis, Funding acquisition, Investigation, Writing - original draft; Qiaohong Wei, Fan Wu, Han Hu, Investigation; Chuan Ma, Lifeng Meng, Xufeng Zhang, Mao Feng, Yu Fang, Data curation, Formal analysis; Olav Rueppell, Formal analysis, Supervision, Writing - review and editing; Jianke Li, Conceptualization, Resources, Supervision, Funding acquisition, Methodology, Project administration

### Author ORCIDs

Bin Han https://orcid.org/0000-0001-6974-8699
Fan Wu http://orcid.org/0000-0001-7923-3808
Olav Rueppell https://orcid.org/0000-0001-5370-4229

### Decision letter and Author response

Decision letter https://doi.org/10.7554/eLife.64830.sa1
Author response https://doi.org/10.7554/eLife.64830.sa2

# Additional files

## Supplementary files

- Supplementary file 1. Statistical differences in sucrose responsiveness of different behavioral phenotypes.
- Supplementary file 2. Neuropeptides identified in the brain of *Apis mellifera ligustica* workers.
- Supplementary file 3. Neuropeptides identified in the brain of *Apis cerana cerana* workers.
- Supplementary file 4. Efficiencies of dsRNA-mediated knockdown of *TRP* and *TRPR*.
- Supplementary file 5. Statistical differences in sucrose responsiveness after injection of ds*GFP*, ds*TRP*, and ds*TRPR*.
- Supplementary file 6. Sequence information of primers used in this study.
- Transparent reporting form

## Data availability

Original data have been deposited to ProteomeXchange Consortium with the dataset identifier PXD018713 under http://proteomecentral.proteomexchange.org or are provided as supplementary data files.

The following dataset was generated:

| Author(s) | Year | Dataset title | Dataset URL | Database and Identifier |
|---|---|---|---|---|
| Han B | 2020 | Honeybee brain neuropeptides LC-MSMS | https://www.ebi.ac.uk/pride/archive/projects/PXD018713 | PRIDE, PXD018713 |

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
