## [Decision Letter]

**Acceptance summary:**

This manuscript identifies Tachykinin signaling in modulating the task-specific behavioral responses in two species of honeybees – *Apis mellifera* and *Apis cerana*. The authors identify tachykinin signaling through a neuropeptidomics approach and verify its role in modulating behavior through an in vivo gain-of-function as well as loss-of-function analysis. They find that Tachykinin signaling reduces the response of bees to their task-specific stimuli, and its absence enhances it.

**Decision letter after peer review:**

Thank you for submitting your article "Regulation of Behavioral Specialization: Tachykinin Inhibits Task-Specific Behavioral Responsiveness in Honeybee Workers" for consideration by *eLife*. Your article has been reviewed by 3 peer reviewers, including Sonia Sen as the Reviewing Editor and Reviewer #1, and the evaluation has been overseen by K VijayRaghavan as the Senior Editor. The following individual involved in review of your submission has agreed to reveal their identity: Olena Riabinina (Reviewer #3).

Essential Revisions:

We have a few major concerns/comments that the authors need to consider, these are listed below:

There were issues with the writing that would help the reader.

Abstract and Introduction: These are written in overly general terms. There are some really nice ideas about behaviour, response threshold, and constraints on them that might be lost on the non-specialist reader. Some specific points that need addressing are:

A rationale for studying neuropeptides and choosing exactly these two species.

Some information on signalling pathway characterised in the cell culture experiments so that experiment are understood more easily.

Support for statements with relevant examples. For example, in line 96 – "two honeybee species" – what species exactly?

Results: This section would be greatly improved if the authors put their findings in context with a motivation and a brief description of the approach taken at the top of each section. For example, what are the implications of the findings described in 2.1? How do they lead the authors to the experiments done in 2.2? It would also help to have a brief concluding summary of finding at the end of each section.

Discussion: In our opinion, a clearer discussion around the function of tachykinin signalling in behavioural specialization in honey bees is needed. Specifically:

Do their results indicate that tachykinin has any causal function in behavioural specialization? If they are suggesting that it does, they might have tested whether changes in tachykinin signaling make nectar foragers to pollen foragers or vice versa? (We are not asking for these experiments, however.)

Do the specific responses of nurse bees, pollen and nectar foragers depend on different levels of tachykinin or a different spatial expression of the tachykinin receptor? Here the authors might discuss published findings.

If "Response thresholds can be modified by biogenic amines, and dopamine, 5-hydroxy-tryptamine, octopamine, and tyramine have been implicated in the regulation of different behaviours of worker bees", what might their functional role be? To be more precise, tyramine and recently leucokinin have been shown to affect sucrose response thresholds in pollen and nectar foragers in similar ways than tachykinin (see: Scheiner et al., 2013; Scheiner et al., 2017; Thamm et al., 2017; Ji et al., 2020). What is their function? The question to be discussed is: Are the response thresholds regulated by one or many neuromodulators, and if so, how might they differ in their function?

The authors should include a discussion of the results of a recently published study that found that the leucokinin receptor is also involved in sugar-response threshold and the division of labor between pollen and nectar foragers in *Apis cerana* (Ji et al., 2020).

We had difficulty with the figures in general. They are very small making it hard to tell the difference between squares, circles and triangles or indeed their colours. The axis labels are also very small. Perhaps the authors could use R.Graphics to improve the figures and make them sharper.

There are also some points related to the figures. These are:

Please plot individual data points as well as the bars (instead of only the bars) in bar graphs.

Figure 1B – how many bees were used here?

There is inconsistency in using abbreviation for bee groups.

Figure 2: is there some better way to show significant differences instead of table next to heatmap?

Include in every figure legend which species and TRP are used

Figure 3: what is log (M)?

We found the biochemistry to be distracting to the story line. The authors could move it into the supplementary figure or condensing it to a single figure at the end of the behavioural manipulation. This would retain focus on the behaviour, which we found to be the most interesting part of this manuscript. They might also want to consider bringing the verification of RNAi up to the main manuscript.

Finally, if possible, the authors could consider exploring the expression of the TRP receptor. Since TRP is likely uniformly distributed, its differential action is likely due to regionalised expression of the receptors. It would be interesting to see if, for example, nurse bees show tachykinin receptor expression in a different group of antennal sensory neurons than pollen and nectar foragers.

---

## [Author Response]

Essential Revisions:We have a few major concerns/comments that the authors need to consider, these are listed below:There were issues with the writing that would help the reader.Abstract and Introduction: These are written in overly general terms. There are some really nice ideas about behaviour, response threshold, and constraints on them that might be lost on the non-specialist reader. Some specific points that need addressing are:A rationale for studying neuropeptides and choosing exactly these two species.

In addition to some other revision to increase the clarity of these sections, we have addressed the two specific points to better explain the rationale for studying neuropeptides and choice of models.

Some information on signalling pathway characterised in the cell culture experiments so that experiment are understood more easily.

We have added some information in the introduction even though TRP signaling was not a-priori assumed to play a role, which makes a detailed introduction into the signaling pathway before presenting the results difficult to rationalize. Furthermore, we agree with the concern that the cell-culture work is not the primary story line and have thus greatly condensed that part in the restructured results.

Support for statements with relevant examples. For example, in line 96 – "two honeybee species" – what species exactly?

We have added this species and tried to improve clarity throughout our text with examples.

Results: This section would be greatly improved if the authors put their findings in context with a motivation and a brief description of the approach taken at the top of each section. For example, what are the implications of the findings described in 2.1? How do they lead the authors to the experiments done in 2.2? It would also help to have a brief concluding summary of finding at the end of each section.

Following this suggestion, we have added introductory and concluding sentences to the Results sections.

Discussion: In our opinion, a clearer discussion around the function of tachykinin signalling in behavioural specialization in honey bees is needed. Specifically:Do their results indicate that tachykinin has any causal function in behavioural specialization? If they are suggesting that it does, they might have tested whether changes in tachykinin signaling make nectar foragers to pollen foragers or vice versa? (We are not asking for these experiments, however.)

In response to this comment, we have developed an additional conceptual figure that is now incorporated into the discussion, along with a text revision to clarify our argument that tachykinin does not necessarily cause the behavioral specialization but instead increases or decreases the specialization.

Do the specific responses of nurse bees, pollen and nectar foragers depend on different levels of tachykinin or a different spatial expression of the tachykinin receptor? Here the authors might discuss published findings.

We have now included the important idea that the spatial expression of the receptor could play a very important role for controlling the specificity of TRP effects in different behavioral phenotypes. However, it is important to note that the overall effect of TRP signaling (to reduce the heightened responsiveness of behavioral specialists to their respective stimuli) is dependent on the overall level of TRP (as indicated by systemic injection and RNAi).

If "Response thresholds can be modified by biogenic amines, and dopamine, 5-hydroxy-tryptamine, octopamine, and tyramine have been implicated in the regulation of different behaviours of worker bees" (line 76-78), what might their functional role be? To be more precise, tyramine and recently leucokinin have been shown to affect sucrose response thresholds in pollen and nectar foragers in similar ways than tachykinin (see: Scheiner et al., 2013; Scheiner et al., 2017; Thamm et al., 2017; Ji et al., 2020). What is their function? The question to be discussed is: Are the response thresholds regulated by one or many neuromodulators, and if so, how might they differ in their function?

Unfortunately, we do not have a sufficient mechanistic understanding to characterize the detailed roles of the biogenic amines and how they relate to neuropeptide action in honey bees. However, our central finding of this study is that TRP signaling seems to decrease (if active) or increase (if inactive) other factors that determine response thresholds. While some connections between response thresholds and biogenic amines or neuropeptides have been demonstrated, the determination of response thresholds is likely complex and we prefer to not include too much speculation about mechanistic interactions. Instead, we provide now a broad conceptual model that highlights the important findings of our study.

The authors should include a discussion of the results of a recently published study that found that the leucokinin receptor is also involved in sugar-response threshold and the division of labor between pollen and nectar foragers in Apis cerana (Ji et al., 2020).

We have added this great study, which was published after our submission and complements our finding about the significance of neuropeptide modulation of behavioral specialization.

We had difficulty with the figures in general. They are very small making it hard to tell the difference between squares, circles and triangles or indeed their colours. The axis labels are also very small. Perhaps the authors could use R.Graphics to improve the figures and make them sharper.

We have redrawn the graphics with larger symbols and axis labels, and provide them now in higher resolution to improve clarity. We hope that these are acceptable, even though “R” might be preferred.

There are also some points related to the figures. These are:Please plot individual data points as well as the bars (instead of only the bars) in bar graphs.

The bar graphs that depict a proportion cannot meaningfully enhanced by plotting individual data points (Figure 1C, 1D and 3C, 3D). We have indicated individual data points for sucrose response score (Figure 1B and 3B). Please note that data are ordinal, not continuous, which makes the distinction of individual data points difficult, even at the best resolution.

Figure 1B – how many bees were used here?

We found indicating the exact numbers in the figure impractical (due to error bars and now individual data points) and therefore have added this information in the figure caption.

There is inconsistency in using abbreviation for bee groups.

We have now taken care to abbreviate the species and behavioral phenotypes consistently throughout the manuscript.

Figure 2: is there some better way to show significant differences instead of table next to heatmap?

We have revised the figure to make the results graphically intuitive.

Include in every figure legend which species and TRP are used

This information is now included.

Figure 3: what is log (M)?

We now clarify that this refers to the logarithm of the molarity of TRP2 (the graph in question is now Figure 4D.

We found the biochemistry to be distracting to the story line. The authors could move it into the supplementary figure or condensing it to a single figure at the end of the behavioural manipulation. This would retain focus on the behaviour, which we found to be the most interesting part of this manuscript. They might also want to consider bringing the verification of RNAi up to the main manuscript.

We have followed this suggestion and greatly reduced the presentation of the biochemical work (now 2.4) and moved it after the TRP2-injection/RNAi experiments (now 2.3). The verification of the RNAi efficacy is included in this section 2.3 (which we now make clearer). The demonstration that TRP2-injection and RNAi of TRP and TRPR affects ERK signaling in-vivo (2.5) represents an independent verification of the biochemical results (that were worked out in cell culture; 2.4) to explain the results from 2.3. Thus, we prefer to keep these experiments/results as the final section (2.5).

Finally, if possible, the authors could consider exploring the expression of the TRP receptor. Since TRP is likely uniformly distributed, its differential action is likely due to regionalised expression of the receptors. It would be interesting to see if, for example, nurse bees show tachykinin receptor expression in a different group of antennal sensory neurons than pollen and nectar foragers.

We agree that the regional characterization of TRPR in the different castes is an interesting next step to be able to come up with precise mechanistic models of how TRP-signaling can affect the responsiveness to different stimuli in different behavioral specialists. However, we consider this a future study that exceeds the scope of this manuscript.